# Leveraging Skills from Unlabeled Prior Data for Efficient Online Exploration

## Abstract

Unsupervised pretraining has been transformative in many supervised domains. However, applying such ideas to reinforcement learning (RL) presents a unique challenge in that fine-tuning does not involve mimicking task-specific data, but rather *exploring* and locating the solution through iterative self-improvement. In this work, we study how unlabeled prior trajectory data can be leveraged to learn efficient exploration strategies. While prior data can be used to pretrain a set of low-level skills, or as additional off-policy data for online RL, it has been unclear how to combine these ideas effectively for online exploration. Our method SUPE (**S**kills from **U**nlabeled **P**rior data for **E**xploration) demonstrates that a careful combination of these ideas compounds their benefits. Our method first extracts low-level skills using a variational autoencoder (VAE), and then *pseudo-relabels* unlabeled trajectories using an optimistic reward model, transforming prior data into high-level, task-relevant examples. Finally, SUPE uses these transformed examples as additional off-policy data for online RL to learn a high-level policy that composes pretrained low-level skills to explore efficiently. We empirically show that SUPE reliably outperforms prior strategies, successfully solving a suite of long-horizon, sparse-reward tasks.

## 1 Introduction

Unsupervised pretraining has been transformative in many supervised domains, such as language (Devlin et al., 2018) and vision (He et al., 2022). Pretrained models can adapt with small numbers of examples, and with better generality (Radford et al., 2019; Brown et al., 2020). However, in contrast to supervised learning, reinforcement learning (RL) presents a unique challenge in that fine-tuning does not involve further mimicking task-specific data, but rather *exploring* and locating the solution through iterative self-improvement. Thus, the key challenge to address in pretraining for RL is not simply to learn good representations, but to learn an effective *exploration strategy* for solving downstream tasks.

Pretraining benefits greatly from the breadth of the data. Unlabeled trajectories (i.e., those collected from previous policies whose objectives are unknown) are the most abundantly available, but using them to solve specific tasks can be difficult. It is not enough to simply copy behaviors, which can differ greatly from the current task. There is an *entanglement* problem – general knowledge of the environment is mixed in with task-specific behaviors. A concrete example is learning from unlabeled locomotion behavior: we wish to learn how to move around the world, but not necessarily to the locations present in the pretraining data. We will revisit this setting in the experimental section.

The entanglement problem can be alleviated through hierarchical decomposition. Specifically, trajectories can be broken into segments of task-agnostic skills, which are composed in various ways to solve various objectives. We posit that unlabeled trajectories thus present a twofold benefit, (1) as a way to learn a diverse set of skills, and (2) as off-policy examples of composing such skills. Notably, prior online RL methods that leverage pretrained skills largely ignore the second benefit, and discard the prior trajectories after the skills are learned (Ajay et al., 2021; Pertsch et al., 2021; Hu et al., 2023; Chen et al., 2024). We instead argue that such trajectories are critical, and can greatly speed up learning. We make use of a simple strategy of learning an optimistic reward model from online samples, and *pseudo-relabeling* past trajectories with an optimistic reward estimate. The past

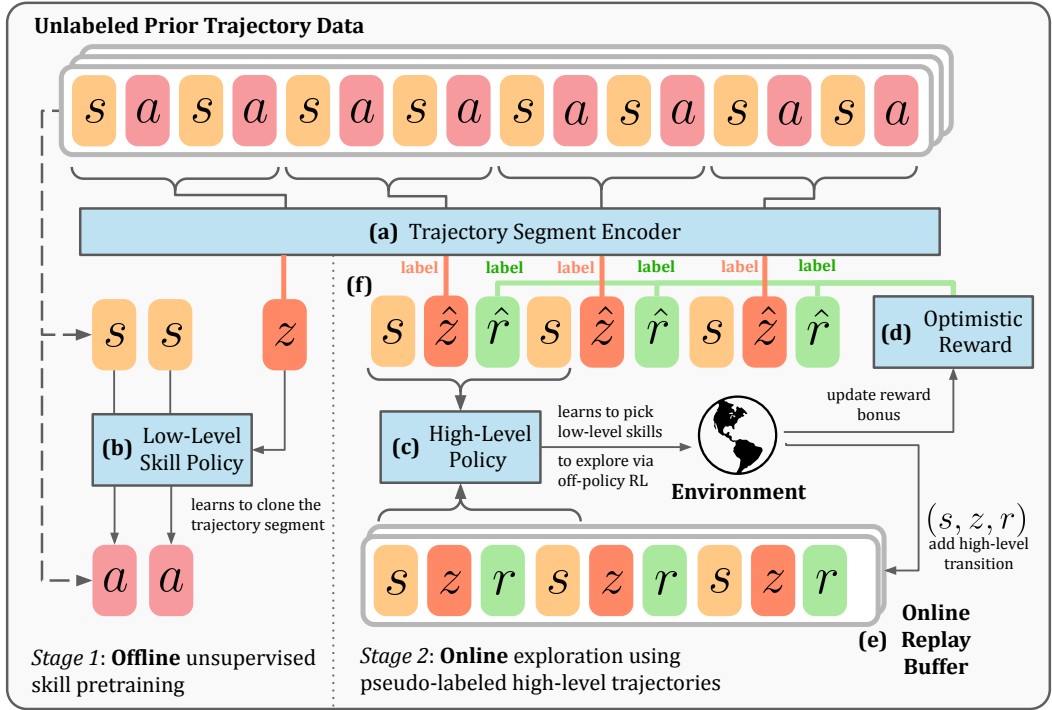

Figure 1: **SUPE utilizes unlabeled trajectory data *twice***, both for *offline* unsupervised skill pretraining and for *online* high-level policy learning using RL. **Left:** in the *offline* pretraining phase *(Stage 1)*, we unsupervisedly learn both a **trajectory segment encoder (a)** and a **low-level latent conditioned skill policy (b)** via a behavior cloning objective where the policy is optimized to reconstruct the action in the trajectory segment. **Right:** in the *online* exploration phase *(Stage 2)*, the pretrained **trajectory segment encoder (a)** and an **optimistic reward module (d)** are used to pseudo-label the prior data and transform it into high-level trajectories **(f)** that can be readily consumed by a high-level off-policy RL agent. Leveraging these offline trajectories and the online replay buffer **(e)**, we learn a **high-level policy (c)** that picks the pretrained low-level skills online to explore in the environment. Finally, the observed transitions and reward values are used to update the optimistic reward module and the online replay buffer.

trajectories can thus be readily utilized as off-policy data, allowing for quick learning even with a very small number of online interactions.

We formalize these insights as SUPE (**S**kills from **U**nlabeled **P**rior data for **E**xploration), a recipe for maximally leveraging unlabeled prior data in the context of exploration. The prior data is utilized in two capacities, the offline and online phases. In the offline pretraining phase, we extract short segments of trajectories and use them to learn a set of low-level skills. In the online phase, we learn a high-level exploration policy, and again utilize the prior data by labelling each trajectory segment with an optimistic reward estimate. By "double-dipping" in this way, we can utilize both the low-level and high-level structure of prior trajectories to enable efficient exploration online.

Our main contribution is a simple method that leverages unlabeled prior trajectory data to both pretrain skills offline and compose these skills efficiently online for exploration. We instantiate SUPE with a variational autoencoder (VAE) to extract low-level skills, and an off-the-shelf off-policy RL algorithm (Ball et al., 2023) to learn a high-level policy from both online and offline data (Figure 1). Our empirical evaluations on a set of challenging sparse reward tasks show that leveraging the unlabeled prior data during both offline and online learning is crucial for efficient exploration, enabling SUPE to find the sparse reward signal more quickly and achieve more efficient learning over all prior methods (none of which are able to utilize the data both online and offline).

## 2 RELATED WORK

**Unsupervised skill discovery.** Unsupervised skill discovery methods first began in the online setting, where RL agents were tasked with learning structured behaviors in the absence of reward

signal (Gregor et al., 2016; Bacon et al., 2017; Florensa et al., 2017; Achiam et al., 2018; Eysenbach et al., 2018; Sharma et al., 2020; Hansen et al., 2020; Park et al., 2023b). These insights naturally transferred to the offline setting as a method of dealing with unlabeled trajectory data. Offline skill discovery methods largely comprise of two categories, those who extract skills based on optimizing unsupervised reward signals (in either the form of policies (Touati et al., 2022; Hu et al., 2023; Frans et al., 2024; Park et al., 2024b) or Q-functions (Chen et al., 2024)), and those who utilize conditional behavior-cloning over subsets of trajectories (Paraschos et al., 2013; Merel et al., 2018; Shankar & Gupta, 2020; Ajay et al., 2021; Singh et al., 2021; Pertsch et al., 2021; Nasiriany et al., 2022). Closest to our method in implementation are Ajay et al. (2021) and Pertsch et al. (2021), who utilize a trajectory-segment VAE to learn low-level skills, and learn a high-level policy online. However, in contrast to prior methods which all utilize offline data purely for skill-learning and do not keep it around during online training, we show that utilizing the data via *relabeling* is critical for fast exploration.

**Offline to online reinforcement learning.** The offline-to-online reinforcement learning methods (Xie et al., 2021b; Song et al., 2023; Lee et al., 2022; Agarwal et al., 2022; Zhang et al., 2023; Zheng et al., 2023; Ball et al., 2023; Nakamoto et al., 2024) focus on efficient online learning with the presence of offline data (often labeled with the reward value). Many offline RL approaches can be applied to this setting – simply run offline RL first on the offline data to convergence as an initialization and then continue training for online learning (using the combined dataset that consists of both offline and online data) (Kumar et al., 2020; Kostrikov et al., 2021; Tarasov et al., 2024). However, such approaches often result in slow online improvements as offline RL objectives tend to overly constrain the policy behaviors to be close to the prior data, limiting the exploration capability. On the other hand, off-policy online RL methods can also be directly applied in this setting by directly treating the offline data as additional off-policy data in the replay buffer and learning the policy from scratch (Lee et al., 2022; Song et al., 2023; Ball et al., 2023). While related in spirit, these methods cannot be directly used in our setting as they require offline data to have reward labels.

**Data-driven exploration.** A common approach for online exploration is to augment reward bonuses to the perceived rewards and optimize the RL agent with respect to the augmented rewards (Stadie et al., 2015; Bellemare et al., 2016; Houthooft et al., 2016; Pathak et al., 2017; Tang et al., 2017; Ostrovski et al., 2017; Achiam & Sastry, 2017; Merel et al., 2018; Burda et al., 2018; Ermolov & Sebe, 2020; Guo et al., 2022; Lobel et al., 2023). While most exploration methods operate in the purely online setting and focus on adding bonuses to the online replay buffer, recent works also start to explore a more data-driven approach that makes use of an unlabeled prior data to guide online exploration. Li et al. (2024) explore adding bonuses to the offline data, allowing them to optimize the RL agent to be optimistic about states in the data, encouraging exploration around the offline data distribution. Our method explores a similar idea of adding bonuses to the offline data but for training a high-level policy, allowing us to compose pretrained skills effectively for exploration. Hu et al. (2023) explore a slightly different strategy of learning a number of policies using offline RL that each optimizes for a random reward function. Then, it samples actions from these policies online to form an action pool from which the online agent can choose to select for exploration. This approach does not utilize offline data during the online phase and require all the policies (for every random reward function) to be represented separately. In contrast, our method makes use of the offline data as off-policy data for updating the high-level policy and our skills are represented using a single network (with the skill latent being the input to our network). As we will show in our experiments, being able to use offline data is crucial for learning to explore in the environment efficiently.

**Hierarchical reinforcement learning.** The ability of RL agents to explore and behave effectively over a long horizon is an important research goal in the field of hierarchical RL (HRL) (Dayan & Hinton, 1992; Dietterich, 2000; Vezhnevets et al., 2016; Daniel et al., 2016a; Kulkarni et al., 2016; Vezhnevets et al., 2017; Peng et al., 2017; Riedmiller et al., 2018; Nachum et al., 2018; Ajay et al., 2021; Shankar & Gupta, 2020; Pertsch et al., 2021; Gehring et al., 2021; Xie et al., 2021a). HRL methods typically learn a high-level policy to leverage a space of low-level primitive policies online. These primitives can be either manually specified (Dalal et al., 2021) or pre-trained using unsupervised skill discovery methods as discussed above. While many existing works learn or fine-tune the primitives online along with the high-level policy (Dietterich, 2000; Kulkarni et al., 2016; Vezhnevets et al., 2016; 2017; Nachum et al., 2018; Shankar & Gupta, 2020), others opt for a less flexible but simpler formulation where the primitives are kept fixed after an initial pre-training phase and only the high-level policy is being learned online (Peng et al., 2017; Riedmiller et al., 2018; Ajay

et al., 2021; Pertsch et al., 2021; Gehring et al., 2021). Our work adopts the later strategy where we offline pre-train skills using a static, unlabeled dataset. None of prior HRL methods simultaneously leverage offline data for skill pre-training and as additional off-policy data for high-level policy learning online. As we show in our experiments, both of them are crucial in achieving sample efficient learning on challenging sparse-reward tasks.

**Options framework.** Many existing works on building hierarchical agents also adopt the options framework (Sutton et al., 1999; Menache et al., 2002; Chentanez et al., 2004; Mannor et al., 2004; Şimşek & Barto, 2004; Şimşek & Barto, 2007; Konidaris, 2011; Daniel et al., 2016a; Srinivas et al., 2016; Daniel et al., 2016b; Fox et al., 2017; Bacon et al., 2017; Kim et al., 2019; Bagaria & Konidaris, 2019; Bagaria et al., 2024). Different from the approach we take that learns latent skills with a fixed time horizon ($H = 4$ in all our experiments), the options framework provides a more flexible way to learn skills with varying time horizon, often defined by learnable initiation and/or termination conditions (Sutton et al., 1999). We opt for the simplified skill definition because it allows us to bypass the need to learn initiation or termination conditions, and frame the skill pretraining phase as a simple supervised learning task.

## 3 PROBLEM FORMULATION

We consider a Markov decision process (MDP) $\mathcal{M} = \{\mathcal{S}, \mathcal{A}, \boldsymbol{P}, \gamma, r, \rho\}$ where $\mathcal{S}$ is the set of all possible states, $\mathcal{A}$ is the set of all possible actions that a policy $\pi(a|s) : \mathcal{S} \mapsto \mathcal{P}(\mathcal{A})$ may take, $\boldsymbol{P}(s'|s, a) : \mathcal{S} \times \mathcal{A} \mapsto \mathcal{P}(\mathcal{S})$ is the transition function that describes the probability distribution over the next state $s'$ given the current state and the action taken at the state, $\gamma$ is the discount factor, $r(s, a) : \mathcal{S} \times \mathcal{A} \mapsto \mathbb{R}$ is the reward function, and $\rho : \mathcal{P}(\mathcal{S})$ is the initial state distribution. We have access to a dataset of transitions that are collected from the same MDP with no reward labels: $\mathcal{D} = \{(s_i, a_i, s'_i)\}$. During online learning, the agent may interact with the environment by taking actions and observes the next state and the reward specified by transition function $\mathcal{P}$ and the reward function $r$. We aim to develop a method that can leverage the dataset $\mathcal{D}$ to efficiently explore in the MDP to collect reward information, and outputs a well-performing policy $\pi(a|s)$ that achieves good cumulative return in the environment $\eta(\pi) = \mathbb{E}_{\{s_0 \sim \rho, a_t \sim \pi(a_t|s_t), s_{t+1} \sim \boldsymbol{P}(\cdot|s_t, a_t)\}} \sum_{t=0}^{\infty} [\gamma^t r(s_t, a_t)]$. Note that this is different from the zero-shot RL setting (Touati et al., 2022) where the reward function is specified for the online evaluation (only unknown during the unsupervised pretraining phase). In our setting, the agent has zero knowledge of the reward function and must actively explore in the environment to identify the task it needs to solve by receiving the reward through environment interactions.

## 4 SKILLS FROM UNLABELED PRIOR DATA FOR EXPLORATION (SUPE)

In this section, we describe in detail how we utilize the unlabeled trajectory dataset to accelerate online exploration. Our method, SUPE, can be roughly divided into two parts. The first part is an offline pretraining phase where we extract skills from the unlabeled prior data with an trajectory-segment VAE. The second part is the online learning phase where we train a high-level off-policy agent to compose the pretrained skills leveraging examples from both prior data and online replay buffer. Algorithm 1 describes our method.

**Pretraining with trajectory VAE.** Since we only have access to an unlabeled dataset of trajectories, we must capture all the behaviors in the data as accurately as possible. At the same time, we aim to make the dataset directly usable for training a high-level skill-setting policy in hope that this high-level policy can be trained in a more sample-efficient way (compared to only having access to the online samples). We achieve this by adopting a trajectory VAE design from prior methods (Ajay et al., 2021; Pertsch et al., 2021) where a short segment of trajectory $\tau = \{s_0, a_0, s_1, \cdots, s_{H-1}, a_{H-1}\}$ is first fed into a trajectory encoder $f_\theta(z|\tau)$ that outputs a distribution over the latent skill $z$, then a skill policy $\pi_\theta(a|s, z)$ is used to reconstruct the actions in the trajectory segment. Such a design helps us directly map trajectory segments to their corresponding skill policies, effectively allowing us to transform the segment into a high-level transition in the form of (*current state:* $s_0$, *action:* $z$, *next state:* $s_H$). As result, such transition can be directly consumed by any off-policy RL algorithm in the online phase to update the high-level policy $\pi_\psi(z|s)$ (as we will explain in the next section in more details). We also learn a state-dependent prior $p_\theta(z|s)$,

---

**Algorithm 1** SUPE

---

1: **Input:** Unlabeled dataset of trajectories $\mathcal{D}$, trajectory segment length $H$ and batch size $B$.
2: **for** each pretraining step **do**
3:     Sample a batch of trajectory segments of length $H$, $\{\tau_1, \cdots, \tau_B\}$ from $\mathcal{D}$
4:     Optimize the skill policy $\pi_\theta(a|s, z)$, the trajectory encoder $f_\theta(z|\tau)$, along with the state-dependent prior $p_\theta(z|s)$ with the VAE loss $\frac{1}{B} \sum_{i=1}^{B} \mathcal{L}_\theta(\tau_i)$ (Equation 1)
5: **end for**
6: $\mathcal{D}_{\text{replay}} \leftarrow \emptyset$
7: Initialize the optimistic reward module $r_{\text{UCB}}(s, z)$ (following (Li et al., 2024))
8: **for** every $H$ online environment steps **do**
9:     Sample the trajectory latent $z \sim \pi_\psi(z|s)$
10:     Run the skill policy $\pi_\theta(a|s, z)$ for $H$ steps in the environment: $\{s_0, a_0, r_0, \cdots, s_H\}$
11:     Add the high-level transition to buffer $\mathcal{D}_{\text{replay}} \leftarrow \mathcal{D}_{\text{replay}} \cup \{(s_0, z, s_H, \sum_{i=0}^{H-1}[\gamma^i r_i])\}$.
12:     Sample a batch of trajectory segments of length $H$, $\{\tau_1, \cdots, \tau_B\}$ from $\mathcal{D}$
13:     Encode each trajectory segment using the trajectory encoder: $\hat{z}^i \sim f_\theta(z|\tau_i)$
14:     Use $f_\theta$ and $r_{\text{UCB}}$ to transform each unlabeled trajectory segment into a high-level transition with pseudo-labels (Equation 2): $\boldsymbol{B}_{\text{offline}} = \{(s_0^i, \hat{z}^i, \hat{r}^i, s_H^i)\}_{i=1}^{B}$
15:     Sample batch $\boldsymbol{B}_{\text{online}}$ from $\mathcal{D}_{\text{replay}}$
16:     Run off-policy RL update on $\boldsymbol{B}_{\text{online}} \cup \boldsymbol{B}_{\text{offline}}$ to train $\pi_\psi(z|s)$.
17: **end for**
18: **Output:** A hierarchical policy consisting of a high-level $\pi_\psi(z|s)$ and low-level $\pi_\theta(a|s, z)$.

---

following the prior works (Pertsch et al., 2021), to help accommodate the difference in behavior diversity of different states. Putting them all together, the loss is shown in Equation 1.

$$\mathcal{L}_\theta(\tau) = \beta D_{\text{KL}}(f_\theta(z|\tau)||p_\theta(z|s_0)) - \mathbb{E}_{z \sim f_\theta(z|\tau)} \left[ \sum_{h=0}^{H-1} \log \pi_\theta(a_h|s_h, z) \right]. \tag{1}$$

**Online exploration with trajectory skills.** Our main goal in the online phase is to learn a high-level off-policy agent that decides which skill to use at a regular interval of $H$ time steps to learn the task quickly. The agent consumes high-level transitions where the state and the next state are separated by $H$ time steps and the action corresponds to the trajectory latent $z$ that is used to retrieve the low-level actions from the skill policy $\pi(a|s, z)$. To make use of the prior data and generate high-level transitions from it, we need both the action and the reward label for each pair of states (that are separated by $H$ steps) in the trajectory. For the action, we can simply sample from the trajectory encoding of the trajectory segment enclosed by the state pair. For the reward, we maintain an upper-confidence bound (UCB) estimate of the reward value for each state and skill pair $(s, z)$ inspired by the prior work (Li et al., 2024) (where it does so directly in the state-action space $(s, a)$), and pseudo-label the transition with such an optimistic reward estimate. The optimistic reward estimate is recomputed before updating the high level agent, since the estimate changes over time, while the trajectory encoding is computed before starting online learning, since this label does not change. The relabeling is summarized below:

$$( \underbrace{s_0}_{\text{state}}, \underbrace{\hat{z} \sim f_\theta(z|\tau)}_{\text{labeled action}}, \underbrace{\hat{r} = r_{\text{UCB}}(s_0, \hat{z})}_{\text{labeled reward}}, \underbrace{s_H}_{\text{next state}} ). \tag{2}$$

**Practical implementation details.** Following prior work on trajectory-segment VAEs (Ajay et al., 2021; Pertsch et al., 2021), we use a Gaussian distribution (with both mean and diagonal covariance learnable) for the trajectory encoder, the skill policy, as well as the state-dependent prior. While Pertsch et al. (2021) use a KL constraint between the high-level policy and the state-dependent prior, we use a simpler design without the KL constraint that works much better (as we show in Appendix F). To achieve this, we adapt the policy parameterization from (Haarnoja et al., 2018), where the action value is enforced to be between $-1$ and $1$ using a $\tanh$ transformation, and entropy regularization is applied on the squashed space. We use this policy parameterization for the high-level policy $\pi(z|s)$ to predict the skill action in the squashed space $z_{\text{sqaushed}}$. We then recover the actual skill action vector by unsquashing it according to $z = \text{arctanh}(z_{\text{sqaushed}})$, so that it can be used by our skill policy $\pi_\theta(a|s, z)$. For upper-confidence bound (optimistic) estimation of the

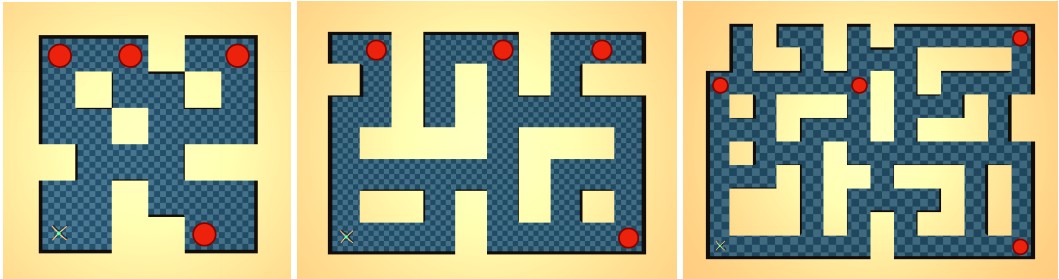

a) **AntMaze**: three maze layouts (`medium`, `large` and `ultra`), and four goals for each layout.

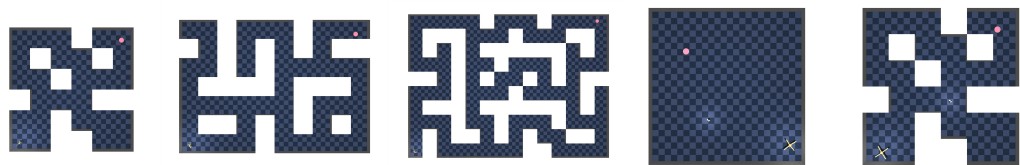

b) **HumanoidMaze**: `medium`, `large`, and `giant`.

c) **AntSoccer**: `arena` and `medium`

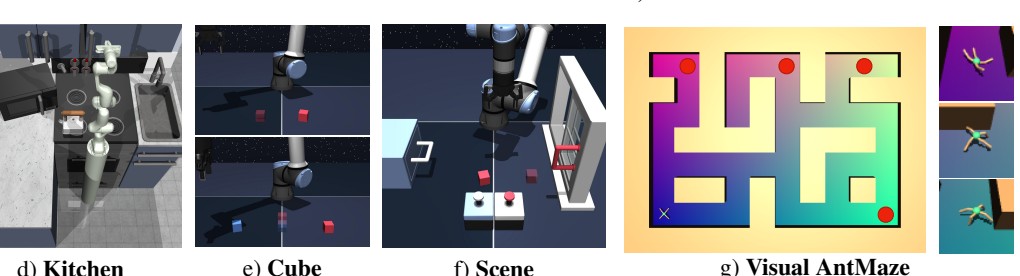

d) **Kitchen**   e) **Cube**   f) **Scene**   g) **Visual AntMaze**

Figure 2: **We experiment on 7 challenging, sparse-reward domains.** *a):* AntMaze with three different layouts and the corresponding four goal locations (denoted as the red dots); *b):* HumanoidMaze with three layouts; *c):* AntSoccer with two layouts *d):* Kitchen; *e):* Cube; *f):* Scene; *g):* Visual AntMaze with colored floor and local $64 \times 64$ image observations.

reward, $(r_{\mathrm{UCB}}(s_0, \hat{z}))$, we directly borrow the UCB estimation implementation in Li et al. (2024) (Section 3, practical implementation section in their paper), where they use a combination of the random network distillation (RND) (Burda et al., 2018) reward bonus and the predicted reward from a reward model (see Appendix C, **Ours** for more details). For the off-policy high-level agent, we follow Li et al. (2024) to use RLPD (Ball et al., 2023) that takes a balanced number of samples from the prior data and the online replay buffer for agent optimization. In addition to using the optimistic offline reward label, we also find that adding the RND reward bonus to the online batch is also helpful to encourage online exploration, so we use it in all our experiments.

## 5 EXPERIMENTAL RESULTS

We present a series of experiments to evaluate the effectiveness of our method to discover fast exploration strategies. We specifically focus on long-horizon, sparse-reward settings, where online exploration is especially important. In particular, we aim to answer the following questions:

1. Can we leverage unsupervised trajectory skills to accelerate online learning?
2. Is our method able to find goals faster than prior methods?
3. How sensitive is the performance of our method is to its hyperparameters?

### 5.1 EXPERIMENTAL SETUP

We conduct our experiments on 7 challenging sparse-reward domains (Figure 2, (a) - (g)). We provide a brief description of each of the domains below with more details available in Appendix D.

**State-based locomotion domains: AntMaze (a), HumanoidMaze (b), AntSoccer (c).** The first set of domains involve controlling and navigating a robotic agent in a complex environment. AntMaze is a standard benchmark for offline-to-online RL from D4RL (Fu et al., 2020). HumanoidMaze and AntSoccer are locomotion domains from OGBench, a offline goal-conditioned RL benchmark (Park et al., 2024a). The goal of agent (either ant or humanoid) is to navigate to a goal location in a fixed maze layout. Each of the AntMaze and HumanoidMaze domains has three different maze layouts. For AntMaze, we test on four different goal location for each maze layout. For HumanoidMaze and AntSoccer we test on one goal, and for HumanoidMaze we use both the `navigate` and `stitch` datasets. In the AntSoccer task, the agent needs to additionally dribble a soccer ball and move the ball to the goal location as well.

**State-based manipulation domains: Kitchen (d), Cube (e), Scene (f).** Next, we consider a set of manipulation domains that require a wide range of manipulation skills. Kitchen is a standard benchmark from D4RL where a robotic arm needs to complete a set of manipulation tasks (e.g., turn on the microwave, move the kettle) in sequence in a kitchen scene. Cube and Scene are two offline goal-conditioned RL benchmark domains from OGBench (Park et al., 2024a). For Cube, the robotic arm must arrange one or more cube objects to desired goal locations (e.g., stacking on top of each other) that mainly involves pick and place motions. For Scene, the robotic arm can interact with a more diverse set of objects: a window, a drawer, a cube and two locks that control the window and the drawer. The tasks in Scene are also relatively longer, requiring a composition of multiple atomic behaviors (e.g., locking and unlocking, opening the drawer/window, moving the cube).

In addition to the six state-based domains, we also experiment with a challenging visual-domain, **Visual AntMaze (g)** introduced by Park et al. (2023a), where the agent must rely on $64 \times 64$ image observations of its surroundings, as well as the proprioceptive information to navigate the maze.

To evaluate our method on these domains, we simply take the datasets in these benchmarks and remove the reward label. We also remove any information in the transition that may reveal the information about the termination of an episode. For all of these tasks, we use a $-1/0$ sparse reward function where the agent receives $-1$ when it has not found the goal and it receives $0$ when it reaches the goal location. For all of the domains above, we use the normalized return, a standard metric for D4RL (Fu et al., 2020) environments, as the main evaluation metric. For the Kitchen domain, the normalized return represents the average percentage of the tasks that are solved. For tasks in other domains, the normalized return represents the average task success rate. For all our figures, the shaded area indicates the standard error and the solid line indicates the mean over random seeds.

## 5.2 COMPARISONS

While there is no existing method in our setting that utilizes unlabeled prior data in both the pretraining phase and the online learning phase, there are methods that use the prior data in either phase. We first consider two baselines that do not use pretraining and directly perform online learning.

**Online.** This baseline discards the offline data and the exploration is done with online reward bonus implemented by random network distillation (RND) (Burda et al., 2018). For all the baselines below (including our method), we add online RND bonus to the replay buffer to encourage exploration.

**ExPLORe (Li et al., 2024).** This baseline is similar to our method in the sense that it also uses exploration bonus and offline data to encourage exploration. The one crucial difference is that it does not perform unsupervised skill pretraining and learns a 1-step policy directly online. As we will show, pretraining is crucial for our method to find goals faster and lead to more efficient online learning. It is worth noting that the original ExPLORe method does not make use of online RND. To make the comparison fair, we additionally add online RND bonus to this baseline to help it explore better online. For completeness, we include the performance of the original ExPLORe in Figure 12.

We then consider additional baselines that use the prior data during a pretraining phase but do not use the data during online learning.

**Diffusion BC + JSRL.** This baseline is an upgraded version of the **BC + JSRL** baseline used in ExPLORe (Li et al., 2024). Instead of using a Gaussian policy (as used by Li et al. (2024)), we use an expressive diffusion model to behavior clone the unlabeled prior data. At the beginning of each online episode, we roll out the policy for a random number of steps from the initial state before using switching to the online RL agent (Uchendu et al., 2023; Li et al., 2023). One might expect

that an expressive enough policy class can model the behavior of the prior good enough such that it can form a good prior for exploration online.

**Online with Skills.** We also consider two skill-based baselines where the prior data is discarded in the online phase and the high-level policy is trained from scratch online with exploration bonus. We experiment with two types of pretraining skills. The first one, is the trajectory VAE skill used in our method. The second one is from a recently proposed unsupervised offline skill discovery method where skills are pretrained to be able to traverse a learned Hilbert representation space (Park et al., 2024b) (HILP). We use the exact same high-level RL agent as our method except that the agent no longer makes use of the prior data online. It is worth noting that the baseline that uses the trajectory VAE skill is very similar to SPiRL (Pertsch et al., 2021), a prior skill-based online RL method that also pretrains skills with a trajectory VAE. The only difference is that we make two additional improvements on top of SPiRL. The first improvement is replacing the KL constraint with entropy regularization (same as our method as described in Section 4, practical implementation details). The second improvement is the online RND bonus that is also added to all other methods.

Finally, we introduce a novel baseline that also uses prior data during pretraining and online exploration, but uses HILP skills rather than trajectory-based skills.

**HILP w/ Offline Data.** We observe that HILP skills can also utilize the offline data via relabeling. Recall that HILP learns a latent space of the observations (via an encoder $\phi_{\text{HILP}}$) and learns skills that move agent in a certain direction $z$ (skill) in the latent space. For any high-level transition $(s_0, s_H)$, we simply take $\hat{z} \leftarrow \frac{\phi_{\text{HILP}}(s_H) - \phi_{\text{HILP}}(s_0)}{\|\phi_{\text{HILP}}(s_H) - \phi_{\text{HILP}}(s_0)\|_2}$, the normalized difference vector that points from $s_0$ to $s_H$ in the latent space. We use the normalized difference vector because the pretrained HILP skill policy takes in normalized skill vectors. We use the exact same high-level RL agent as our method except that the skill relabeling is done by computing the latent difference rather than using the trajectory encoder ($f_\theta(z|\tau)$).

For the visual antmaze environment, we use the same image encoder used in RLPD (Ball et al., 2023). We also follow one of our baselines, ExPLORe (Li et al., 2024), to use ICVF (Ghosh et al., 2023), a method that uses task-agnostic value functions to learn image/state representations from passive data. ICVF takes in an offline unlabeled trajectory dataset with image observations and pretrain an image encoder in an unsupervised manner. Following ExPLORe, we take the weights of the image encoder from ICVF pretraining to initialize the image encoder's weights in the RND network. To make the comparison fair, we also apply ICVF to all baselines (details in Appendix E).

### 5.3 CAN WE LEVERAGE UNSUPERVISED TRAJECTORY SKILLS TO ACCELERATE ONLINE LEARNING?

Figure 3 shows the aggregated performance of our approach on all seven domains. Our method outperforms all prior methods on domains except Scene, where our novel baseline **HILP with Offline Data** performs slightly better. It is worth noting that **HILP with Offline Data** also leverages offline data twice (one of the key ideas behind our method), both during offline and online learning. Both HILP-based methods (**Online with HILP Skills** and **HILP with Offline Data**) perform well on Scene, Single Cube, and AntSoccer, but struggle to learn on other tasks. The **Online with Trajectory Skills** baseline also consistently performs worse than our method across all seven domains, which demonstrates the importance of using prior data for online learning of the high-level policy, since that is only difference between this baseline and **Ours**. **ExPLORe** uses offline data during online learning, but does not pretrain skills, leading to slower learning on all seven environments and difficulty achieving any significant return on any domains other than the easier AntMaze and Single Cube tasks. We also report the performance on individual AntMaze mazes and Kitchen tasks in Figure 11, and observe that our method outperforms the baselines more on harder environments. This trend continues with HumanoidMaze, where individual results in Figure 16 show that **Ours** is the only method to achieve nonzero final return on the more difficult large and giant mazes. These experiments suggest that pretraining skills and the ability to leverage the prior data are both crucial for achieving efficient online learning, especially in more challenging environments. For the visual domain, we additionally perform an ablation study to assess the importance of the ICVF pretrained representation, which we include in Appendix E. While ICVF combines synergistically with our method to further accelerate learning and exploration, initializing RND image encoder weights using ICVF is not critical to its success.

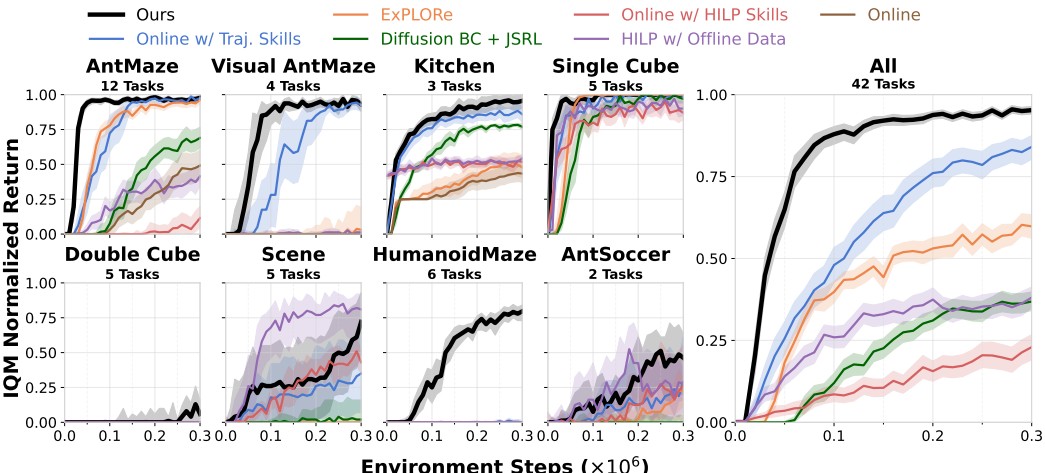

Figure 3: **Aggregated normalized return across seven different domains (Single-Cube and Double-Cube are two sub-domains of Cube). Ours** achieves the best performance on all domains except on Scene where **HILP w/ Offline Data** achieves better performance. **HILP w/ Offline Data** is a novel baseline that we introduce which also uses the offline data both during offline skill pre-training and online learning. Section 5.2 contains details on the baselines we compare with. We omit the **Online** baseline on the harder domains (Cube, Scene, HumanoidMaze, and AntSoccer) as it is consistently worst than other methods. For Kitchen, we use 16 seeds. For AntMaze, Visual AntMaze, HumanoidMaze, and AntSoccer, we use 8 seeds. For the rest, we use 4 seeds. All of the aggregated plots use the interquantile mean (IQM) metric following Agarwal et al. (2021) with 95% stratified bootstrap confidence intervals.

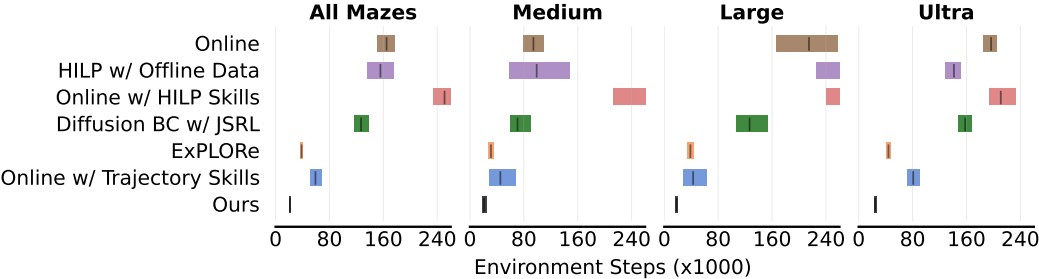

Figure 4: **Interquartile mean (IQM) of the number of environment steps taken to reach the goal (smaller the better).** The first goal time is considered to be $300 \times 10^3$ steps if the agent never finds the goal. Our method is the most consistent, achieving performance better than all other baselines on all layouts (4 tasks/goals for each layout and 8 seeds for each task). The plot is generated using the *rliable* library with 95% stratified bootstrap confidence intervals (Agarwal et al., 2021).

## 5.4 IS OUR METHOD ABLE TO FIND GOALS FASTER THAN PRIOR METHODS?

Even though we have demonstrated that our method is able to achieve higher success rate faster than prior works, it is still not clear if our method can actually lead to better exploration (instead of simply learning the high-level policy better). In this section, we study the exploration aspect in isolation in the AntMaze domain. Figure 4 reports the number of online environment interaction steps for the agent to reach the goal for the first time. Such a metric allows us to assess how efficiently the agent explores in the maze whereas the success rate metric only measures how good the agent is at reaching the desired goal. It is possible for an agent to be good at exploration, but bad at reaching goals consistently and vice-versa. Figure 4 shows that our method reaches goals faster than every baseline on all three maze layouts, which confirms that our method not only learns faster, but does so by exploring more efficiently. For completeness, Table 3 reports the same metric but for each individual goal location and maze layout, and we also include the success rate for each maze layout and goal location in Appendix I.2.

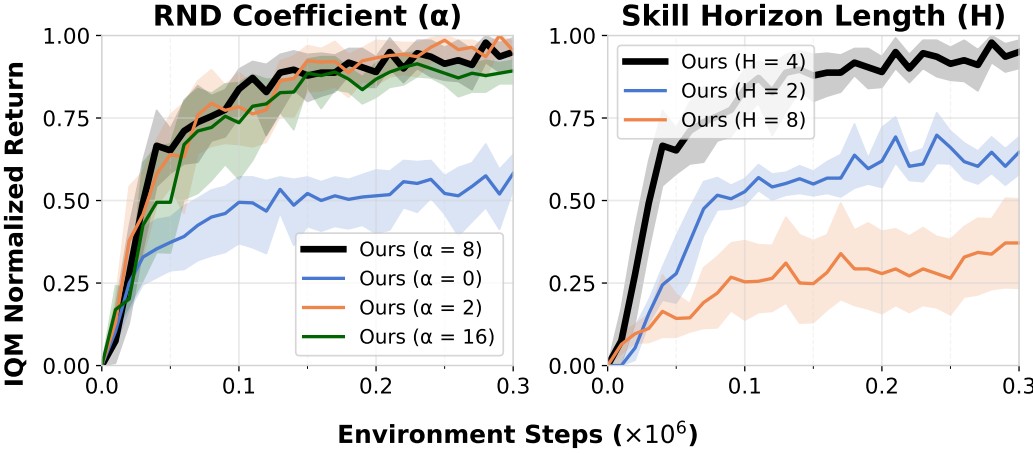

Figure 5: **Sensitivity analysis on the RND coefficient ($\alpha$) and the skill horizon length ($H$) on a subset of tasks.** We report the interquartile mean (IQM) of the normalized return across seven domains (we select one representative task per domain) for different hyperparameter values. The performance of our method is not very sensitive to the magnitude of $\alpha$ as long as it is within a reasonable range $(2, 16)$ (**Ours** uses $\alpha = 8$). Without the bonus ($\alpha = 0$), our method performs significantly worse. A skill horizon length of 4 performs significantly better than a horizon length of 2 or 8. We use a skill length of 4 for all skill experiments. The normalized return for each individual task we use for this analysis can be found in Appendix H.

### 5.5 How sensitive is the performance of our method is to its hyperparameters?

In this section, we study the sensitivity of two hyperparameters, 1) $\alpha$: the amount of optimism (RND coefficient) used when labeling offline data with UCB rewards (Equation 2), and 2) $H$: the length of the skill. We select one representative task from each domain and study the how different $\alpha$ and $H$ values affect our performance on these tasks (Figure 5). For the RND coefficient, we test a wide range of values. When $\alpha = 0$, the RND exploration bonus is turned off, it significantly lowers the performance of our method, highlighting the importance of optimistic labeling on efficient online exploration. While we observe some variability across individual tasks (Appendix 5, Figure 9), our method is generally not very sensitive to the RND coefficient value. The aggregated performance is similar for $\alpha \in \{2, 4, 8, 16\}$. Another key hyperparmeter in our method is the skill horizon length (see Figure 5, right). We find that while there is some variability across individual tasks (Appendix 5, Figure 10), a skill horizon length of 4 generally performs the best, and that shorter or longer horizons perform much worse on certain tasks. We use $H = 4$ for all experiments in the paper.

## 6 Discussion and Limitations

In this work, we propose a novel method, SUPE, that leverages unlabeled prior trajectory data to accelerate online exploration and learning. The key insight is to use unlabeled trajectories *twice*, to 1) extract a set of low-level skills offline, and 2) serve as additional data for a high-level off-policy RL agent to compose these skills to explore in the environment. This allows us to effectively combine the strengths from unsupervised skill pretraining and sample-efficient online RL methods to solve a series of challenging long-horizon sparse reward tasks significantly more efficiently than existing methods. Our work opens up avenues in making full use of prior data for scalable, online RL algorithms. First, our pre-trained skills remain frozen during online learning, which may hinder online learning when the skills are not learned well or need to be updated as the learning progresses. Such problems could be alleviated by utilizing a better skill pretraning method, or allowing the low-level skills to be fine-tuned online. A second limitation of our approach is the reliance on RND to maintain an upper confidence bound on the optimistic reward estimate. Although we find that RND works without ICVF on high-dimensional image observations in Visual AntMaze, the use of RND in other high dimensional environments may require more careful consideration. Possible future directions include examining alternative methods of maintaining this bound.

## 7 REPRODUCIBILITY STATEMENT

We include all the implementation details in Appendix B and C, and we include the code in the supplementary material along with the commands to reproduce all the experiments in our paper.

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

## A    Compute Resources

We run all our experiments on NVIDIA A5000 GPU and V100 GPUs.

We first calculate required compute for AntMaze experiments. For each maze-goal configuration, each of our pretraining run takes about two hours and online training also takes about two hours each. To reproduce the results of our methods, it requires 8 (seeds) $\times 3$ (maze layouts) $\times (1$ (pretraining) $+ 4$ (online learning) $)\times 2$ (hours per run) $= 240$ GPU hours. We have eight baselines that take similar GPU hours, which bring the total estimated GPU hours required to be around 2160.

The runtime per Kitchen experiment is similar. There are only 3 environments, but we do 16 seeds. This means we need about 16 (seeds)$\times 3$ (Kitchen tasks) $\times (1$ (pretraining) $+1$ (online learning) $)\times 2$ (hours per run) $= 192$ GPU hours for our method. We only train six baselines (we do not run Ex-PLORe (No Online RND) ablation, or KL ablation), so this gives 1344 hours. We also calculate the compute used for the Visual AntMaze experiments. On average it takes approximately 8 hours to train a pretraining checkpoint and approximately 24 hours to do online learning. This means it requires 8 (seeds) $\times (1$ (pretraining) $\times 8$ (hours per run)$+4$ (online learning) $\times 24$ (hours per run)) $= 832$ hours for our method. We have six baselines in the main figure. Additionally, we add 4 additional baselines in the ICVF ablation. This gives a total of 9152 hours for the Visual AntMaze results.

For computing the runtime of remaining experiments on OGBench environments, we approximate skill checkpoint as taking 4 hours to train, and running online to take about 4 hours as well. For the 15 manipulation tasks, we train 4 seeds per tasks, for the locomotion tasks, we train 8 seeds. We have 2 methods to pretrain (Traj. skills and HILP), and need to train checkpoints for each Humanoid dataset (6), AntSoccer maze (2), and manipulation task suite (3). This means 8 (seeds) $\times$ 4 (hours per run) $\times 2$ (methods) $\times 8$ (tasks) $+ 4$ (seeds) $\times 4$ (hours per run) $\times 2$ (methods) $\times 3$ (tasks) $= 608$ hours. We have 15 tasks (manipulation) trained with four seeds per task online, 8 (locomotion) trained with eight seeds, and six methods. This means 8 (seeds) $\times 4$ (hours per run)$\times 6$ (methods) $\times 8$ (tasks) $+4$ (seeds) $\times 4$ (hours per run) $\times 6$ (methods) $\times 15$ (tasks) $= 2976$ hours. This gives a total compute of 3585 hours for OGBench results.

For the ablations on other AntMaze environments, including the play dataset and dataset quality ablation, each pretraining and online run took about 3 hours. We did four seeds, and 6 environments. This gives approximately 4 (seeds) $\times$ (1 (pretraining) $+ 1$ (online learning) ) $\times$ 3 (hours per run) $\times 6$ (mazes) $\times 4$ (methods) $= 576$ hours. We also did an RND ablation, were we evaluated 6 additional RND coefficients with 8 seeds on one goal, which requires 8 (seeds) $\times 3$ (hours per run)$\times 6$ (coefficients) $= 144$ hours, as well as a horizon ablation test which used approximately 4 (seeds) $\times$ (1 (pretraining) $+ 1$ (online learning) ) $\times$ 3 (hours per run) $= 24$ hours. Ground truth experiments used a similar amount of time, with 2 methods that did not require additional pretraining and 4 seeds, giving 24 more hours. This gives about 768 hours of other ablations on AntMaze.

Next, we look at the compute required for the data ablation experiments. These are additional AntMaze experiments on just the goals in `antmaze-large` maze. Thus, we have approximately 8 (seeds) $\times$ 1 (maze layouts) $\times$ 2 (data ablations) $\times$ (1 (pretraining) $+ 4$ (online learning) ) $\times$ 2 (hours per run) $= 160$ GPU hours to reproduce our method. We include 5 additional baselines, bringing the total compute for data ablations to 960 GPU hours.

Thus, in total the results in this paper required approximately $16,625$ GPU hours, or about $1.9$ GPU years. Note this is an approximate upper bound, since not all methods required training checkpoints, and checkpoints were shared between different baselines that both uses trajectory skills or both used HILP skills.

## B    VAE Architecture and hyperparameters

We use a VAE implementation from Park et al. (2024b). The authors kindly shared with us their OPAL implementation (which produces the results of the OPAL baseline in the paper). In this implementation, the VAE encoder is a recurrent neural network that uses gated-recurrent units (Cho et al., 2014) (GRU). It takes in a short sequence of states and actions, and produces a probabilistic

output of a latent $z$. The reconstruction policy decoder is a fully-connected network with ReLU activation (Nair & Hinton, 2010) that takes in both the state in the sequence as well as the latent $z$ to output an action distribution.

| Parameter Name | Value |
|---|---|
| Batch size | 256 |
| Optimizer | Adam |
| Learning rate | $3 \times 10^{-4}$ |
| GRU Hidden Size | 256 (AntMaze, Kitchen, VisualAntMaze) |
| | 512 (AntSoccer, HumanoidMaze, Scene, Cube) |
| GRU Layers | 2 hidden layers (AntMaze, Kitchen, VisualAntMaze) |
| | 3 hidden layers (AntSoccer, HumanoidMaze, Scene, Cube) |
| KL Coefficient ($\beta$) | 0.1 (AntMaze, HumanoidMaze, Kitchen, VisualAntmaze, AntSoccer) |
| | 0.2 (Cube, Scene) |
| VAE Prior | state-conditioned isotropic Gaussian distribution over the latent |
| VAE Posterior | isotropic Gaussian distribution over the latent |
| Reconstruction Policy Decoder | isotropic Gaussian distribution over the action space |
| Latent Dimension | 8 |
| Trajectory Segment Length ($H$) | 4 |
| Image Encoder Latent Dim | 50 |

Table 1: VAE training details.

In the online phase, our high-level policy is a Soft-Actor-Critic (SAC) agent (Haarnoja et al., 2018) with 10 critic networks, entropy backup disabled and LayerNorm added to the critics following the architecture design used in RLPD (Ball et al., 2023). We follow a similar strategy in ExPLORe (Li et al., 2024) where we sample 128 offline samples and 128 online samples and add RND reward bonus to all of the samples. The main difference is in the original ExPLORe paper is that they only add reward bonus to the offline data as additionally adding the bonus to the online replay buffer does not help for the maze goals they tested. In our experiments, we add the reward bonus to both offline data and online data, as it leads to better performance in goals where there is limited offline data coverage (see Appendix I.2).

| Parameter | Value |
|---|---|
| Batch size | 256 |
| Discount factor ($\gamma$) | 0.99 (AntMaze, VisualAntmaze, Kitchen) |
| | 0.995 (HumanoidMaze, Cube, Scene, AntSoccer) |
| Optimizer | Adam |
| Learning rate | $3 \times 10^{-4}$ |
| Critic ensemble size | 10 |
| Critic minimum ensemble size | 1 for all methods on AntMaze HumanoidMaze, AntSoccer; |
| | 2 for all methods on Kitchen, Scene, Cube; |
| | 1 for non-skill based methods on Visual AntMaze, |
| | 2 for skill-based methods on Visual AntMaze. |
| UTD Ratio | 20 for state-based domains, 40 for Visual AntMaze |
| Actor Delay | 20 |
| Network Width | 256 (AntMaze, Kitchen, Visual AntMaze) |
| | 512 (HumanoidMaze, AntSoccer, Cube, Scene) |
| Network Depth | 3 hidden layers. |
| Initial Entropy Temperature | 1.0 on Kitchen, 0.05 on all other environments |
| Target Entropy | $-\dim(\mathcal{A})/2$ |
| Entropy Backups | False |
| Start Training | after 5K env steps (RND update starts after 10K steps) |
| RND coefficient ($\alpha$) | 2.0 for non-skill based, 8.0 for skill based methods |

Table 2: Hyperparameters for the online RL agent following RLPD (Ball et al., 2023)/ExPLORe (Li et al., 2024). For Diffusion BC + JSRL on AntMaze, we use an initial entropy temperature of 1.0 because it works much better than 0.05. We use a $4\times$ larger RND coefficient in skill-based methods such that the reward bonus we get for each step in the skill horizon stays roughly proportional to the non-skill-based methods.

## C  IMPLEMENTATION DETAILS FOR BASELINES

**Diffusion BC + JSRL.** We use the diffusion model implementation from (Hansen-Estruch et al., 2023). Following the paper's implementation, we train the model for 3 million gradient steps with a dropout rate of 0.1 and a cosine decaying learning rate schedule from the learning rate of 0.0003. In the online phase, in the beginning of every episode, with probability $p$, we rollout the diffusion policy for a random number of steps that follows a geometric distribution $\text{Geom}(1 - \gamma)$ before sampling actions from the online agent (inspired by (Li et al., 2023)). A RND bonus is also added to the online batch on the fly with a coefficient of 2.0 to further encourage online exploration of the SAC agent. The same coefficient is used in all other non-skill based baselines. For skill based baselines, we scale up the RND coefficient by the horizon length (4) to account for different reward scale. Following the **BC + JSRL** baseline used in ExPLORe (Li et al., 2024), we use a SAC agent with an ensemble of 10 critic networks, one actor network, with no entropy backup and LayerNorm in the critic networks. This configuration is used for all baselines on all environments. On AntMaze, We perform a hyperparameter sweep on both $p = \{0.5, 0.75, 0.9\}$ and the geometric distribution parameter $\gamma = \{0.99, 0.995, 0.997\}$ on the large maze with the top right goal and find that $p = 0.9$ and $\gamma = 0.99$ works the best. We also use these parameters for the Visual AntMaze experiments. On AntSoccer, we use the same $p$ but raise the discount $\gamma$ to match the environment discount rate of 0.995. On HumanoidMaze, we perform a sweep over $p = \{0.5, 0.75, 0.9\}$ on `humanoidmaze-medium-navigate-v0` and find that $p = 0.75$ works best. We still use $\gamma = 0.995$. On Scene and Cube, we do a similar sweep for $p$ on the first task of Scene, Single Cube, and Double Cube, and find that $p = 0.5$ works best. We still use $\gamma = 0.995$. For Kitchen, we perform a sweep on the parameter $p = \{0.2, 0.5, 0.75, 0.9\}$ and find that 0.75 works best. We use $\gamma = 0.99$. We take the minimum of one random critic for AntMaze, Visual AntMaze, AntSoccer, and HumanoidMaze, and the minimum of two random critics for Kitchen, Scene, and Cube. For all methods, we use the same image encoder used in RLPD (Ball et al., 2023) for the Visual AntMaze task, with a latent dimension of 50 (encoded image is a 50 dimensional vector), which is then concatenated with proprioceptive state observations.

**ExPLORe.** We directly use the open-source implementation from `https://github.com/facebookresearch/ExPLORe/`. The only difference we make is to adjust the RND coefficient from 1.0 to 2.0 and additionally add such bonus to the online replay buffer (the original method only adds to the offline data). Empirically, we find a slightly higher RND coefficient improves performance slightly. The SAC configuration is the same as that of the Diffusion BC + JSRL agent. We found that taking the minimum of one random critic on Visual AntMaze worked better for **ExPLORe** than taking the minimum of two, so all non-skill based baselines use this hyperparameter value.

**Online RL with trajectory skills.** This baseline is essentially our method but without using the trajectory encoder in the VAE to label trajectory segments (with high-level skill action labels), so all stated implementation decisions also apply to **Ours**. Instead, we treat it directly as a high-level RL problem with the low-level skill policy completely frozen. The SAC agent is the same as the previous agents, except for on Visual AntMaze, where taking the minimum of 2 critics from the ensemble leads to better performance for **Ours**, so we use this parameter setting for all skill-based benchmarks. We compute the high-level reward as the discounted sum of the rewards received every $H$ environment steps. During the $5 \times 10^3$ steps before the start of training, we sample random actions from the state-based prior. For Visual AntMaze, we use the learned image encoder from the VAE to initialize both the critic image encoder and the RND network. If using ICVF, we initialize the RND network with the ICVF encoder instead.

**Online RL with HILP skills.** This baseline is the same as the one above but with the skills from a recent unsupervised offline skill discovery method, HILP (Park et al., 2024b). We use the official open-source implementation `https://github.com/seohongpark/HILP` and run the pre-training to obtain the skill policies. Then, we freeze the skill policies and learn a high-level RL agent to select skills every $H$ steps.

**HILP w/ offline data.** This novel baseline is the same as **Online w/ HILP Skills**, except that we also relabel the offline trajectories and use them as additional data for learning the high-level policy online (similar to our proposed method). To relabel trajectories with the estimated HILP skill, we compute the difference in the latent representation of the final state $s_H$ and initial state $s_0$ in the

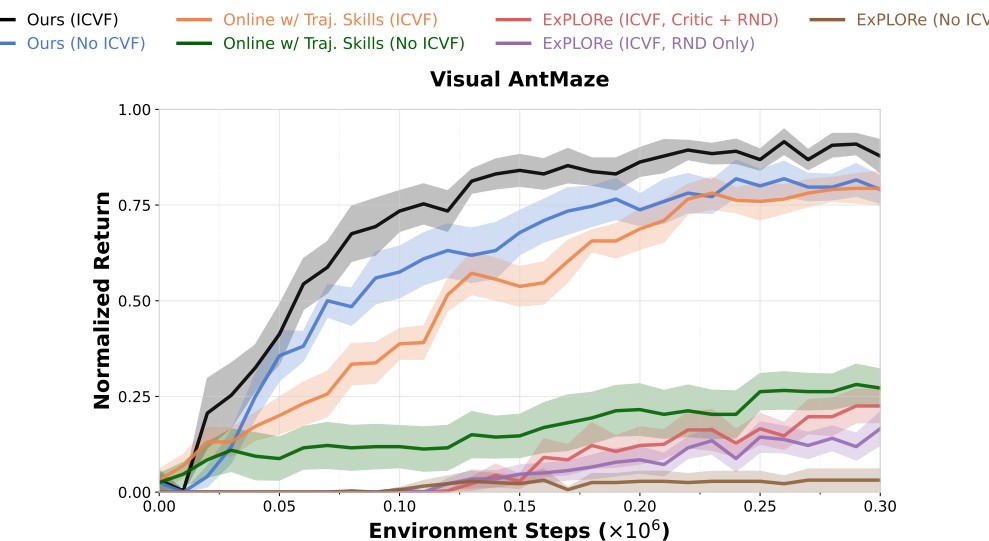

Figure 6: **Success rate on Visual AntMaze environment with and without ICVF. Ours** works well without ICVF, almost matching the original performance. However, the other baselines **Online w/ Trajectory Skills** and **ExPLORe** achieve far worse performance without ICVF, which shows that using offline data both for extracting skills and online learning leads to better utilization of noisy exploration bonuses. Initializing ExPLORe critic with ICVF helps, but does not substantially change performance.

trajectory, so $\hat{z} \leftarrow \frac{\phi_{\text{HILP}}(s_H) - \phi_{\text{HILP}}(s_0)}{\|\phi_{\text{HILP}}(s_H) - \phi_{\text{HILP}}(s_0)\|_2}$. We normalize the skill vector since the pretrained HILP policies use a normalized vector as input. The high-level RL agent is the same as our method, except the skill relabeling is done using the latent difference rather than the trajectory encoder.

**Ours.** We follow Li et al. (2024) (ExPLORe) to relabel offline data with optimistic reward estimates using RND and a reward model. For completeness, we describe the details below. We initialize two networks $g_\phi(s, z)$, $\bar{g}(s, z)$ that each outputs an $L$-dimensional feature vector predicted from the state and (tanh-squashed) high-level action. During online learning, $\bar{g}(s, z)$ is fixed and we only update the parameters of the other network $g_\phi(s, z)$ to minimize the $L_2$ distance between the feature vectors predicted by the two networks on the new high-level transition $(s_0^{\text{new}}, z^{\text{new}}, r^{\text{new}}, s_H^{\text{new}})$:

$$\mathcal{L}(\phi) = \|g_\phi(s_0^{\text{new}}, z^{\text{new}}) - \bar{g}(s_0^{\text{new}}, z^{\text{new}})\|_2^2.$$

In addition to the two networks, we also learn a reward model $r_\psi(s, z)$ that minimizes the reward loss below on the transitions $(s_0, z, r, s_H)$ from online replay buffer:

$$\mathcal{L}(\psi) = \|r_\psi(s_0, z) - r\|_2^2.$$

We then form an optimistic estimate of the reward value for the offline data as follows:

$$r_{\text{UCB}}(s, z) \leftarrow r_\psi(s_0, z) + \alpha \|g_\phi(s_0, z) - \bar{g}(s_0, z)\|_2^2,$$

where $\alpha$ controls the strength of the exploration tendency (RND coefficient). For AntMaze environments (both state-based and visual), we find that it is sufficient to use the minimum reward, $-1$, to label the offline data without a performance drop, so we opt for such a simpler design for our experiments.

## D  DOMAIN DETAILS

**D4RL AntMaze with Additional Goal Locations.** D4RL AntMaze is a standard benchmark for offline-to-online RL (Fu et al., 2020; Ball et al., 2023) where an ant robot needs to navigate around a maze to a specified goal location. We benchmark on three mazes of increasing size, `antmaze-medium`, `antmaze-large`, and `antmaze-ultra`. We take the D4RL dataset for medium and large mazes as well as the dataset from Jiang et al. (2022) for the ultra maze (we use the `diverse` version of the datasets for all these layouts). We then remove the termination and

reward information from the dataset such that the agent does not know about the goal location a priori. For each of the medium, large, ultra mazes, we test with four different goal locations that are hidden from the agent. See Figure 2a for a visualization of the mazes and the four goal locations that we use for each of them. We use a $-1/0$ sparse reward function where the agent receives $-1$ when it has not found the goal and it recieves $0$ when it reaches the goal location and the episode terminates. The ant always starts from the left bottom corner of the maze, and the goal for the RL agent is to reach the goal consistently from the start location. It is worth noting that the goal is not known a priori and the offline data is unlabeled. In order to learn to navigate to the goal consistently, the agent first needs to traverse in the maze to gather information about where the goal is located.

**D4RL Kitchen** is another standard benchmark for offline-to-online RL (Fu et al., 2020; Nakamoto et al., 2024), where a Franka robot arm is controlled to interact with various objects in a simulated kitchen scenario. The desired goal is to complete four tasks (open the microwave, move the kettle, flip the light switch, and slide open the cabinet door) in sequence. In the process, the agent attains a reward equal to the number of currently solved tasks. The benchmark contains three datasets, `kitchen-mixed`, `kitchen-partial`, and `kitchen-complete`. `kitchen-complete` is the easiest dataset, which only has demonstrations of the four tasks completed in order. `kitchen-partial` adds additional tasks, but the four tasks are sometimes completed in sequence. `kitchen-mixed` is the hardest, where the four tasks are never completed in sequence. To adapt this benchmark in our work, we remove all the reward labels in the offline dataset.

We also include four additional domains repurposed from a goal-conditioned offline RL benchmark (Park et al., 2024a).

**OGBench HumanoidMaze** is a navigation task similar to AntMaze, but with the Ant agent replaced by a Humanoid agent. HumanoidMaze is much more challenging than AntMaze because Humanoid control is much harder with much higher action dimensionality (21 vs. 8). We benchmark on all six datasets in the benchmark. They involve three mazes of increasing size, `humanoid-medium`, `humanoid-large`, and `humanoid-giant`, and two types of datasets, `navigate` (collected by noisy expert policy that randomly navigates around the maze) and `stitch` (containing only short segments that test the algorithm's ability to stitch them together).

**OGBench AntSoccer** is a navigation task similar to AntMaze, but with the added complexity where the Ant agent must first travel to the location of a soccer ball, then dribble the soccer ball to the goal location. We benchmark on the two `navigate` datasets, `antsoccer-arena-navigate` and `antsoccer-medium-navigate`.

**OGBench Cube and Scene** are two manipulation domains with a range of manipulation tasks for each domain. Cube involves using a robot arm to arrange blocks from an initial state to a goal state, which requires pick and place actions to move and/or stack blocks. We benchmark on the two sub-domains, `cube-single` and `cube-double` which include one and two blocks, respectively. We also benchmark on the Scene environment with two buttons, a drawer, and a window. The most difficult task requires the agent to perform 8 atomic actions: unlock the drawer and window, open drawer, pick up block, place block in drawer, close drawer, open window, lock drawer and window. We benchmark on all 5 tasks for `cube-single`, `cube-double`, and `scene`, and use the `play` datasets collected by non-Markovian expert policies with temporally correlated noise.

Aside from the state-based domains above, we also consider a visual domain below to test the ability of our method in scaling up to high-dimensional image observations.

**Visual AntMaze** is a benchmark introduced by Park et al. (2023a), where the agent must rely on $64 \times 64$ image observations of its surroundings, as well as proprioceptive information including body joint positions and velocities to navigate the maze. In particular, the image is the only way for the agent to locate itself within the maze, so successfully learning to extract location information from the image is necessary for successful navigation. The floor is colored such that any image can uniquely identify a position in the maze. The maze layout is the same as the large layout in the state-based D4RL AntMaze benchmark above and we also use the same additional goals. The reward function and the termination condition are also the same as the state-based benchmark.

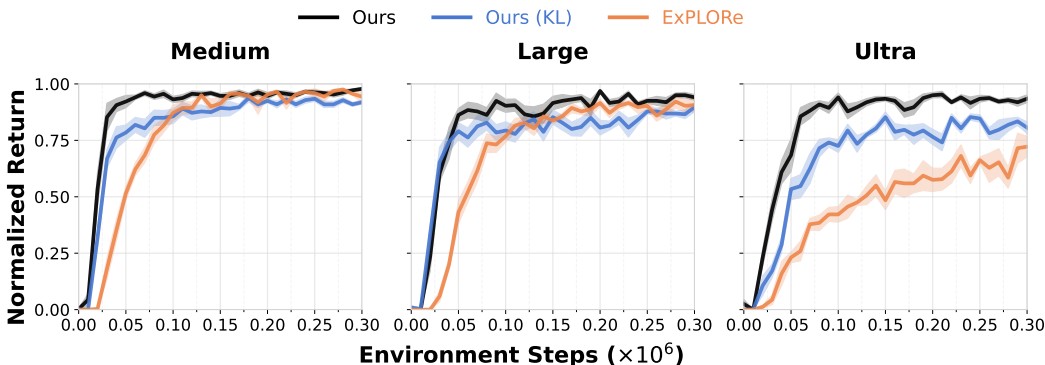

Figure 7: **Normalized return on three AntMaze mazes, comparing Ours with a KL regularized alternative (Ours (KL)).** We that **Ours** consistently outperforms **Ours (KL)** on all three mazes, with initial learning that is at least as fast and significantly improved asymptotic performance. Only **Ours** is able to meet or surpass the asymptotic performance of **ExPLORe** on all mazes.

## E  ICVF IMPLEMENTATION DETAILS AND ABLATION EXPERIMENTS FOR VISUAL ANTMAZE

We use the public implementation from the authors of (Ghosh et al., 2023) at `https://github.com/dibyaghosh/icvf_release` and run the ICVF training for $75,000$ gradient steps to obtain the pre-trained encoder weights, following (Li et al., 2024). Then, we initialize the encoder of the RND network with these weights before online learning. It is worth noting that this is slightly different from the prior work (Li et al., 2024) that initializes both the RND network and the critic network. In Figure 6, we examine the performance of **Ours**, **Online w/ Trajectory Skills**, and **ExPLORe** with and without ICVF. Both of the baselines perform much better with the ICVF initialization, suggesting that ICVF might play an important role in providing more informative exploration signal. **Ours**, without using ICVF, can already outperform the baselines with ICVF. By both extracting skills from offline data and training with offline data, we are able to learn better from less informative exploration signals. We also observe that initializing the critic with ICVF (as done in the original paper (Li et al., 2024)) helps improve the performance of **ExPLORe** some, but does not substantially change performance.

## F  KL PENALTY ABLATION

In Figure 7, we compare the performance of **Ours** with a version of our method that uses a KL-divergence penalty with the state-based prior (as used in a previous skill-based method (Pertsch et al., 2021)), **Ours (KL)**. In **Ours**, as discussed in Section 4, we borrow the policy parameterization from Haarnoja et al. (2018) and adopt a tanh policy parameterization with entropy regularization on the squashed space. Pertsch et al. (2021) parameterize the higher level policy as a normal distribution and is explicitly constrained to a learned state-dependent prior using a KL-divergence penalty, with a temperature parameter that is auto-tuned to match some target value by using dual gradient descent on the temperature parameter. They do not use entropy regularization. Keeping everything else about our method the same, we instantiate this alternative policy parameterization in **Ours (KL)**. We sweep over possible target KL-divergence values $(5, 10, 20, 50)$ and initial values for the temperature parameter $(100, 1, 0.1)$ using the performance on `antmaze-large`, but find that these parameters do not substantially alter performance. As shown in Figure 7, **Ours** performs at least as well as **Ours (KL)** in the initial learning phase, and has better asymptotic performance on all three mazes, matching or beating **ExPLORe**, on all three mazes. It seems likely that not having entropy regularization makes it difficult to appropriately explore online, and that explicitly constraining to the prior may prevent further optimization of the policy. Attempts at combining an entropy bonus and KL-penalty lead to instability and difficulty tuning two separate temperature parameters. Additionally, in the Kitchen domain, the KL objective is unstable, since at some states the prior standard deviation is quite small, leading to numerical instability. In contrast, adopting the

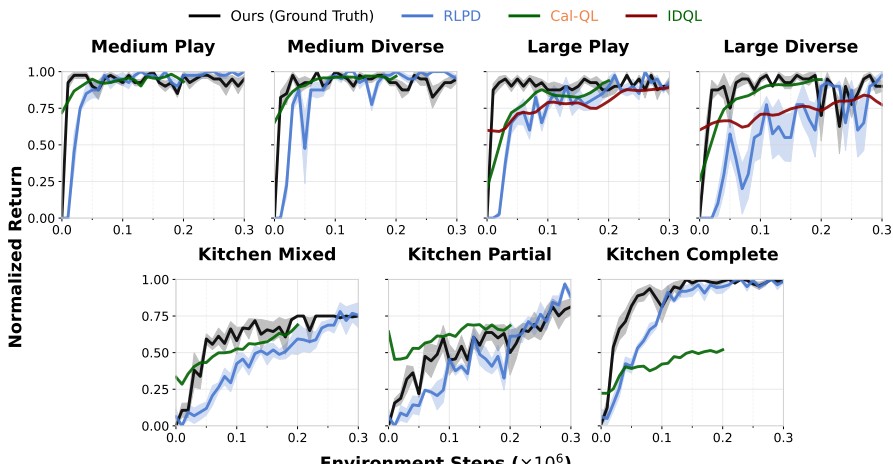

Figure 8: **Comparison to best methods in the offline-to-online RL setting.** Our method with the ground truth offline reward outperforms state-of-the-art offline-to-online methods such as Cal-QL (Nakamoto et al., 2024) and IDQL (Hansen-Estruch et al., 2023) in six of the seven tasks. Both the curves of Cal-QL and IDQL are directly taken from their papers. For **Ours (Ground Truth)** and **RLPD**, 4 seeds are used and standard error is shown.

tanh policy parameterization from Haarnoja et al. (2018) is simple, performs better, and encounters none of these issues in our experiments.

## G    COMPARISON WITH OFFLINE-TO-ONLINE RL METHODS

Our setting is different from the typical offline-to-online RL setting since we do not have reward label in the offline data. However, our method can be easily adapted to the offline-to-online RL setting by simply removing the online reward prediction model and replacing the reward prediction (of the offline transitions) with the ground truth reward. As shown in Figure 8, on the four AntMaze tasks (top right goal), our method with ground truth outperforms all baselines considered (CalQL (Nakamoto et al., 2024), RLPD (Ball et al., 2023), IDQL (Hansen-Estruch et al., 2023)). In Kitchen, our method with ground truth outperforms all baselines on the `kitchen-mixed` and `kitchen-complete` datasets, but performs slightly worse than CalQL on `kitchen-partial`.

## H    SENSITIVITY ANALYSIS

We picked one representative task from each domain: AntMaze Large top right goal, Kitchen Mixed, HumanoidMaze Giant Stitch, Single Cube Task 1, Double Cube Task 1, Scene Task 1, and AntSoccer Medium Navigate. Figure 9 shows the performance of our method with different RND coefficients. We see that an RND coeffcient of zero is very bad for performance on the locomotion tasks, and that the best nonzero RND coeffcient varies between environments. For all experiments in the paper, we selected the middle value of eight, and kept it the same across all domains. Figure 10 shows the performance of our method with different skill horizon lengths. We see that while a horizon length of 2 attains better final performance on AntMaze Large Top Right at the cost of slower initial exploration, it performs worse than a horizon length of 4 on all other environments. A longer horizon length of 8 seems like it could be slightly better in Kitchen Mixed and Scene, but performs much worse in all other tasks. We found that using a horizon length of 4 generally worked well on all tasks, so we used this length for all trajectory-skill based experiments in this paper.

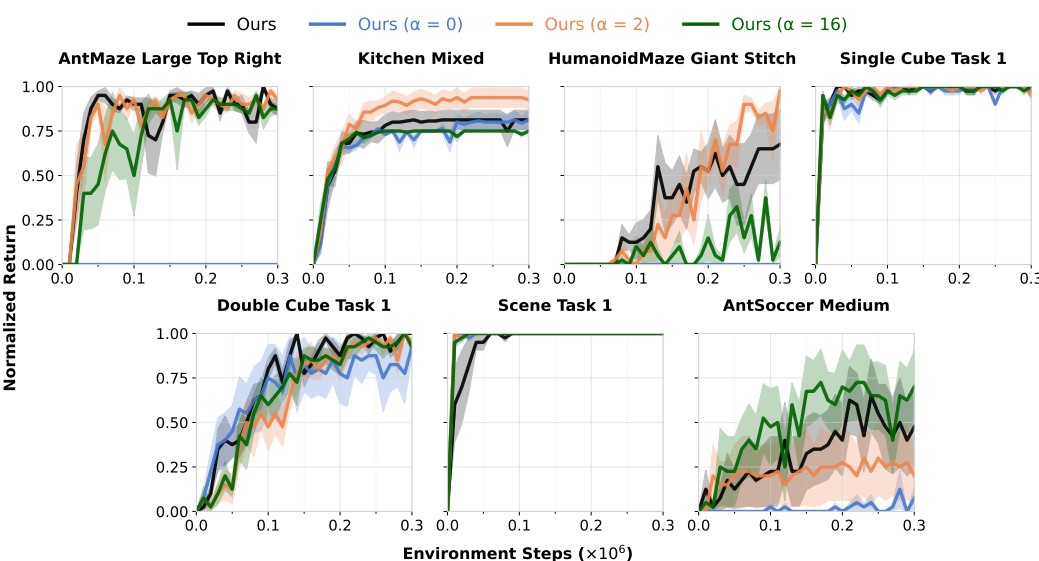

Figure 9: **Sensitivity analysis for the RND coefficient.** RND is essential to strong performance on the lomotion tasks (AntMaze, AntSoccer, HumanoidMaze). The best coefficient varies between tasks. We use the middle value of 8 for all experiments in this paper, scaled accordingly based on horizon length. All curves use 4 seeds, and standard error is shown.

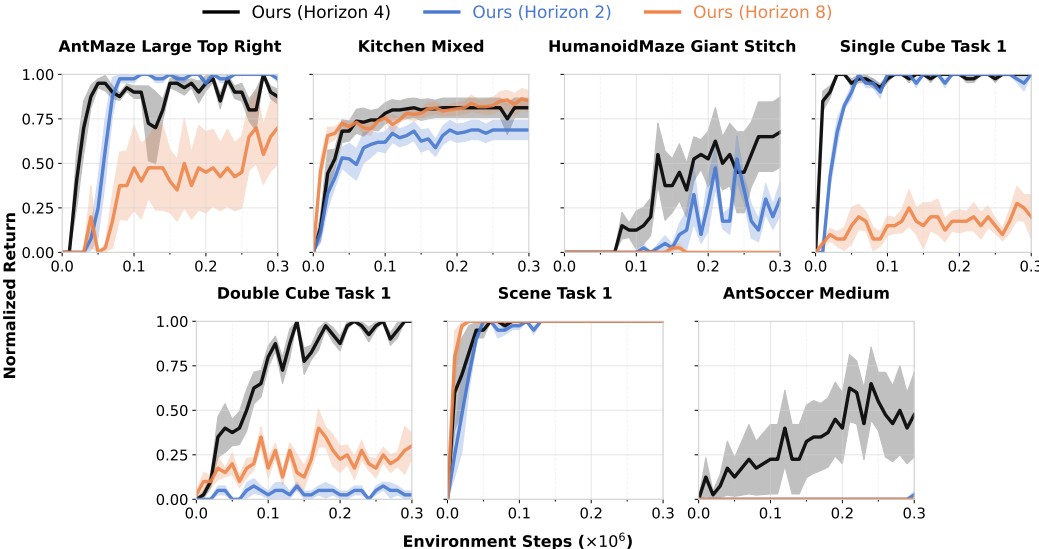

Figure 10: **Sensitivity analysis for the skill horizon length.** A horizon length of 4 generally performs the best across all environments. A horizon length of 2 performs relatively well in AntMaze Large Top Right, but performs poorly in all other environments. A longer horizon length of 8 performs slightly better than a horizon length of 4 in Kitchen Mixed and Scene Task 1, but performs poorly in all other tasks. We used a horizon length of 4 for all experiments in this paper. All curves use 4 seeds, and standard error is shown.

# I   STATE-BASED D4RL RESULTS

In this section, we summarize our experimental results on two state-based D4RL domains: AntMaze and Kitchen. Figure 11 shows the comparison of our method against all the baselines.

| Maze Layout | Goal Location | Methods without Pretraining | | Methods with Pretraining | | | | |
| --- | --- | --- | --- | --- | --- | --- | --- | --- |
| | | *Online + RND* | *ExPLORe* | *Diffusion BC w/ JSRL* | *Online w/ Trajectory Skills* | *Online w/ HILP Skills* | *HILP w/ Offline Data* | *Ours* |
| **Medium** | Top Left | $71 \pm 5.0$ | $27 \pm 3.2$ | $60 \pm 8.1$ | $\mathbf{21 \pm 4.1}$ | $120 \pm 47$ | $27 \pm 6.2$ | $\mathbf{14 \pm 3.1}$ |
| | Top Right | $100 \pm 16$ | $\mathbf{29 \pm 2.8}$ | $85 \pm 19$ | $76 \pm 26$ | $160 \pm 40$ | $\mathbf{72 \pm 36}$ | $\mathbf{22 \pm 3.2}$ |
| | Bottom Right | $230 \pm 38$ | $35 \pm 4.9$ | $99 \pm 15$ | $77 \pm 34$ | $300 \pm 0$ | $270 \pm 33$ | $\mathbf{22 \pm 4.4}$ |
| | Center | $210 \pm 32$ | $71 \pm 8.0$ | $260 \pm 28$ | $26 \pm 3.4$ | $260 \pm 28$ | $300 \pm 0.0$ | $\mathbf{18 \pm 1.7}$ |
| | *Aggregated* | $150 \pm 14$ | $40 \pm 2.0$ | $130 \pm 10$ | $50 \pm 11$ | $210 \pm 17$ | $170 \pm 13$ | $\mathbf{19 \pm 1.8}$ |
| **Large** | Top Left | $72 \pm 10$ | $33 \pm 2.9$ | $52 \pm 3.3$ | $\mathbf{22 \pm 4.2}$ | $300 \pm 0$ | $300 \pm 0.0$ | $\mathbf{21 \pm 2.8}$ |
| | Top Right | $220 \pm 20$ | $49 \pm 7.7$ | $220 \pm 28$ | $190 \pm 27$ | $280 \pm 20$ | $110 \pm 36$ | $\mathbf{27 \pm 2.6}$ |
| | Bottom Right | $280 \pm 15$ | $34 \pm 1.8$ | $160 \pm 22$ | $140 \pm 22$ | $280 \pm 21$ | $260 \pm 19$ | $\mathbf{21 \pm 1.8}$ |
| | Top Center | $220 \pm 28$ | $\mathbf{48 \pm 5.2}$ | $120 \pm 8.8$ | $\mathbf{59 \pm 12}$ | $240 \pm 23$ | $\mathbf{33 \pm 8.5}$ | $\mathbf{39 \pm 6.2}$ |
| | *Aggregated* | $200 \pm 8.9$ | $41 \pm 2.7$ | $140 \pm 13$ | $100 \pm 13$ | $270 \pm 12$ | $180 \pm 13$ | $\mathbf{27 \pm 1.7}$ |
| **Ultra** | Top Left | $76 \pm 7.0$ | $34 \pm 4.9$ | $91 \pm 11$ | $36 \pm 11$ | $39 \pm 21$ | $\mathbf{15 \pm 5.3}$ | $\mathbf{17 \pm 3.6}$ |
| | Top Right | $300 \pm 0.0$ | $92 \pm 20$ | $290 \pm 7.8$ | $120 \pm 14$ | $260 \pm 19$ | $150 \pm 32$ | $\mathbf{37 \pm 5.5}$ |
| | Bottom Right | $300 \pm 0.0$ | $70 \pm 8.0$ | $300 \pm 0.0$ | $130 \pm 16$ | $240 \pm 28$ | $67 \pm 12$ | $\mathbf{34 \pm 6.0}$ |
| | Top Center | $230 \pm 35$ | $29 \pm 5.5$ | $230 \pm 29$ | $\mathbf{75 \pm 16}$ | $100 \pm 32$ | $\mathbf{17 \pm 1.7}$ | $\mathbf{22 \pm 4.4}$ |
| | *Aggregated* | $230 \pm 9.3$ | $56 \pm 5.4$ | $230 \pm 9.1$ | $90 \pm 7.1$ | $160 \pm 14$ | $61 \pm 8.2$ | $\mathbf{27 \pm 2.4}$ |
| **Aggregated** | | $190 \pm 6.9$ | $46 \pm 2.3$ | $160 \pm 6.3$ | $80 \pm 5.9$ | $210 \pm 11$ | $130 \pm 7.4$ | $\mathbf{25 \pm 1.4}$ |

Table 3: **The number of environment steps ($\times 10^3$) taken before the agent find the goal.** Lower is better. The first goal time is considered to be $300 \times 10^3$ steps if the agent never finds the goal. We see that our method is the most consistent, achieving performance **as good as or better than all other methods** in each of the 4 goals across 3 different maze layouts. The error quantity indicated is standard error over 8 seeds. The method that has the lowest mean is in bold and all the other methods with values that are not statistically significantly higher are also in bold. We used the $t$-test with $p = 0.05$ to determine the statistical significance.

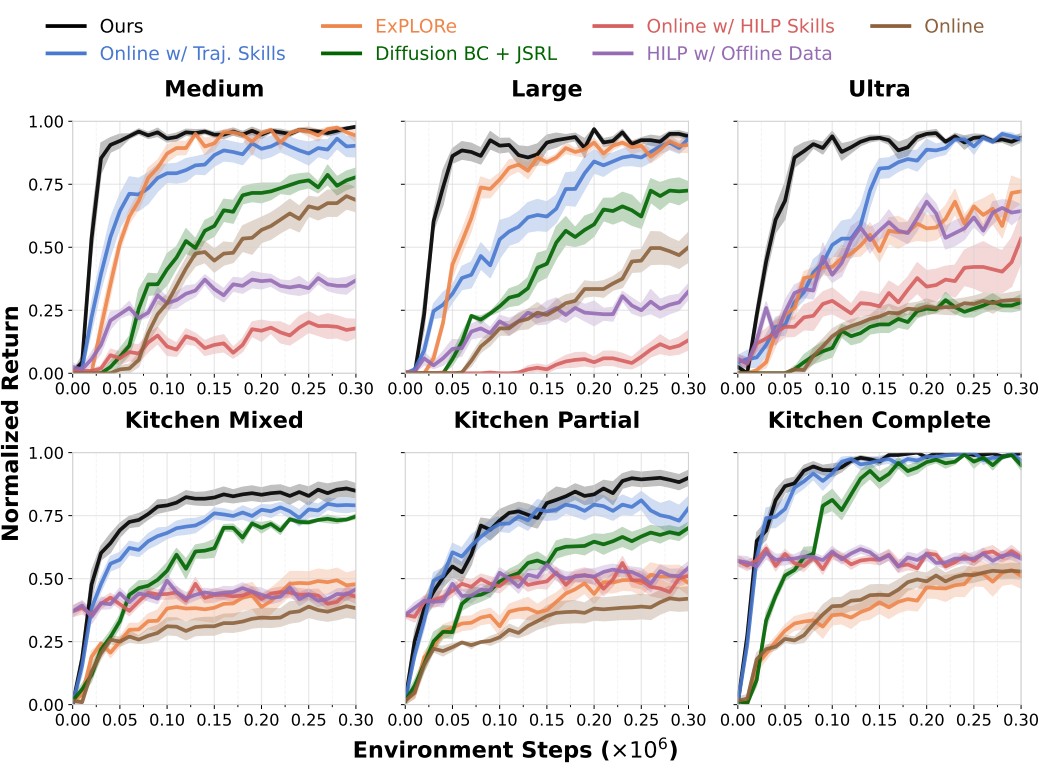

Figure 11: **Normalized return on individual AntMaze and Kitchen tasks. Ours** achieves the strongest performance on all tasks. **Online w/ Trajectory Skills** learns much slower on all AntMaze tasks, and is asymptotically worse on two of the three more challenging Kitchen tasks. **ExPLORe** struggles to learn on Kitchen, and performs worse as maze size increases. None of the other baselines are competitive on any tasks. Each curve is an average over four goals with 8 seeds for AntMaze, and 16 seeds for Kitchen.

## I.1 EXPLORATION EFFICIENCY

Figure 14 shows the percentage of the maze that the agent has covered throughout the training. The coverage of skill-based methods that do not use prior data during online learning, **Online w/ Trajec-**

**tory Skills** and **Online w/ HILP Skills**, significantly lags behind baselines that use offline data after 50,000 environment steps. Many methods achieve similar coverage on `antmaze-medium`, likely because the maze is too small to differentiate the different methods. **Ours** is able to achieve the highest coverage on the `antmaze-ultra`, and is only surpassed on `antmaze-large` by **HILP w/ Offline Data**, which has high first goal times and slow learning. Thus, the coverage difference can likely be at least partially attributed to **HILP w/ Offline Data** struggling to find the goal and continuing to explore after finding the goal. All non-skill based methods struggle to get competitive coverage levels on `antmaze-large` and `antmaze-ultra`. This suggests both pretraining skills and the ability to leverage prior data online are crucial for efficient exploration, and our method effectively compounds their benefits.

### I.2    FULL D4RL ANTMAZE RESULTS

We evaluate the success rate of the our algorithm compared to the same baseline suite as in the main results section for each individual goal and maze layout and report the results in Figure 12. We also include **ExPLORe** both with and without an online RND bonus. Online RND helps **ExPLORe** the most for the `antmaze-medium` bottom-right goal, where there is sparse offline data coverage for a considerable radius around the goal. We hypothesize that with the absence of online RND, the agent is encouraged to only stay close to the offline dataset, making it more difficult to find goals in less well-covered regions. On the flip side, for some other goals with better offline data coverage, like the `antmaze-large` top-right goal, online RND can make the performance worse. For every goal location, **Ours** consistently matches or outperforms all other methods throughout the training process.

We also evaluate the coverage at every goal location for every method for each maze layout and show the result in Figure 13. The coverage varies from goal location to goal location as some goal locations are harder to reach. Generally, the agent stops exploring once it has learned to reach the goal. **Ours** consistently has the best initial coverage for 11 out of 12 goals, though sometimes has lower coverage compared to other methods later in training. However, this is likely due in large part to successfully learning how to reach that goal quickly, and thus not exploring further.

### I.3    D4RL PLAY DATASET

Since there is limited performance difference between the diverse and the play datasets, we only report the performance on the diverse datasets. For completeness, we also include the results of the play datasets in Figure 15. The results on the play datasets are consistent with our results in the main body of the paper where our method outperforms all baseline approaches consistently with better sample efficiency.

## J    OGBENCH RESULTS

In this section, we include the full results on individual tasks for each of the four OGBench domains (HumanoidMaze, Cube, Scene, and AntSoccer) (Park et al., 2024a).

### J.1    HUMANOIDMAZE

As shown in Figure 16, our method substantially outperforms all prior methods on the difficult HumanoidMaze environment. It is the only method to achieve nonzero return on the more difficult `large` and `giant` mazes, and performs approximately four times better than the next best baseline, **Online w/ Traj. Skills**, on the `medium` environments. These results show that on difficult, long horizon tasks, using offline data during online learning is essential for strong exploration performance.

### J.2    CUBE AND SCENE

As shown in Figure 17, **Ours** matches or outperforms the next best baseline on 11 of the 15 tasks. The novel baseline that we introduce **HILP w/ Offline Data** which also uses offline data for skill

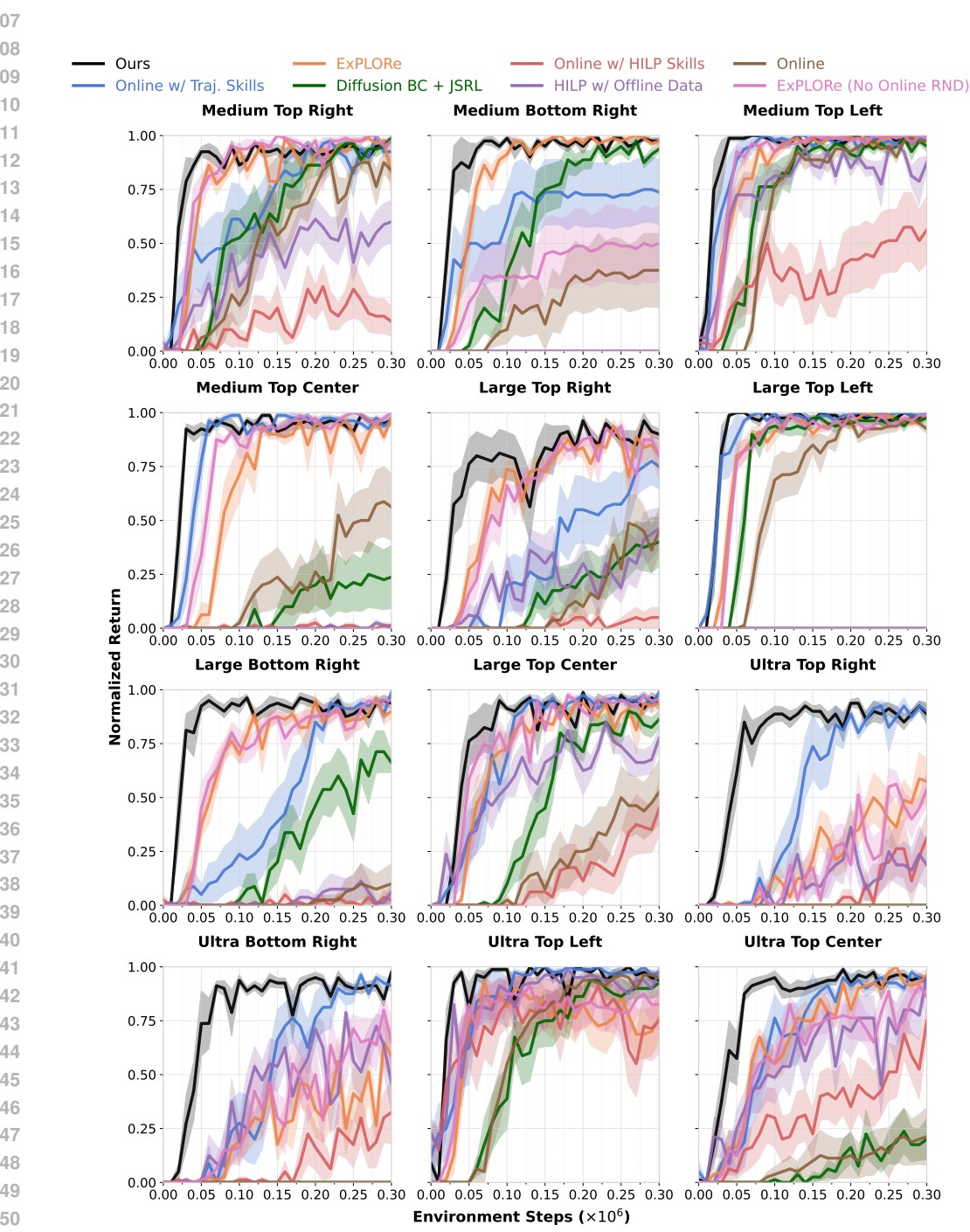

Figure 12: **Success rate by goal location.** The addition of online RND in **ExPLORe** leads to better performance on goals with less offline data coverage, and slightly worse performance on goals well-represented in the dataset. **Ours** consistently matches are outperforms all other methods on all goals throughout training.

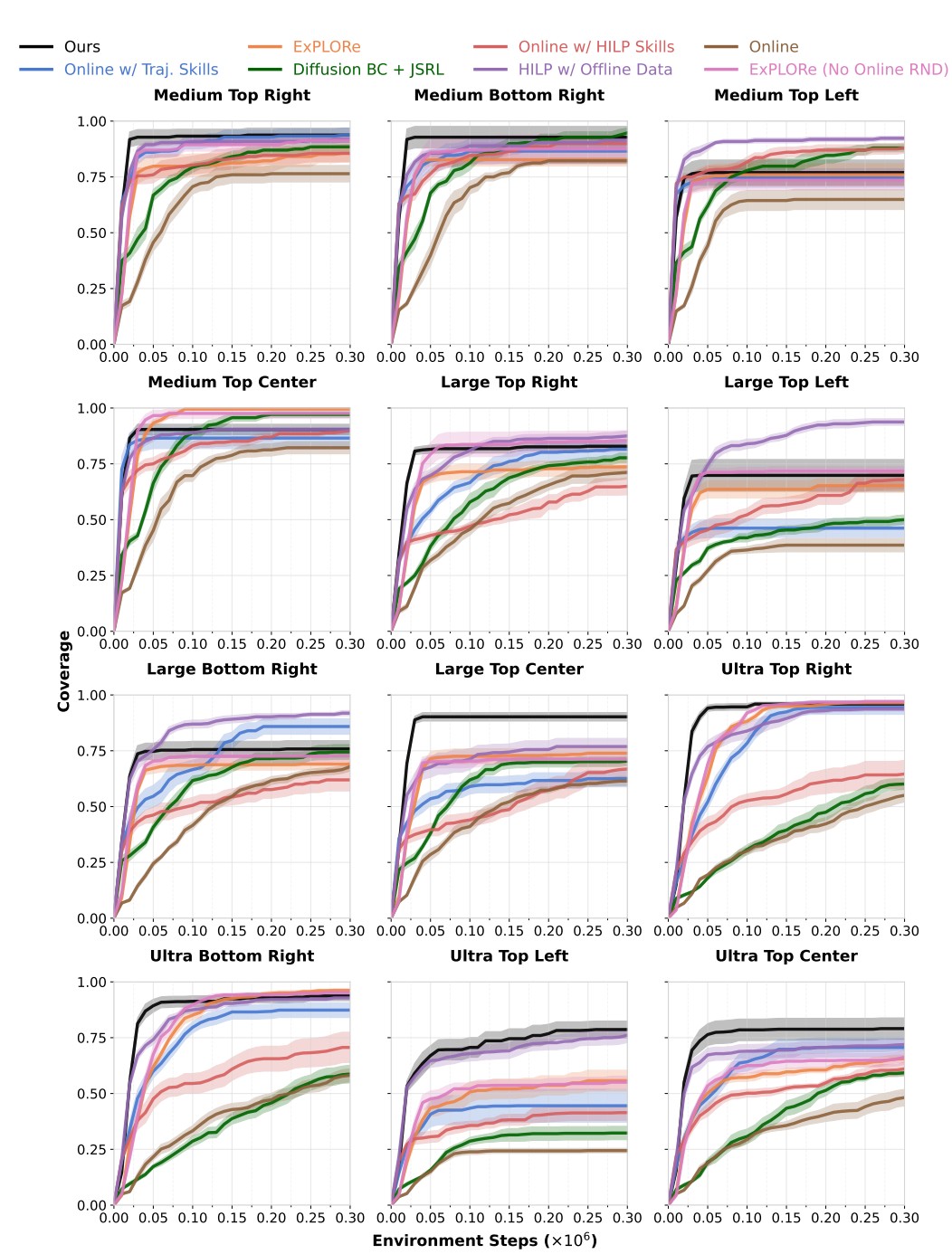

Figure 13: **Coverage for every goal location on three antmaze environments.** There is significant variation between goals, and **Ours** consistently has the best initial coverage performance on 11 of 12 goals. Flattening coverage compared to other methods can be at least partially attributed to having already found the goal, and sucessfully optimizing reaching that goal, rather than continuing to explore after already finding the goal.

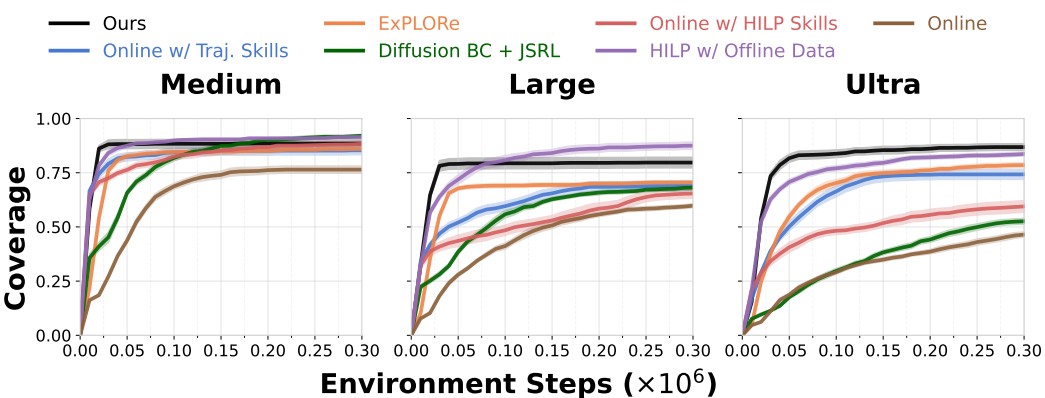

Figure 14: **Coverage on three different AntMaze mazes, averaged over runs on four goals. Ours** has the best coverage performance on the challenging `antmaze-ultra`, and is only passed by **HILP w/ Offline Data** on `antmaze-large`. **Online w/ Traj. Skills** and **Online with HILP Skills** struggle to explore after initial learning, and **Online** and **Diffusion BC + JSRL** generally perform poorly at all time steps.

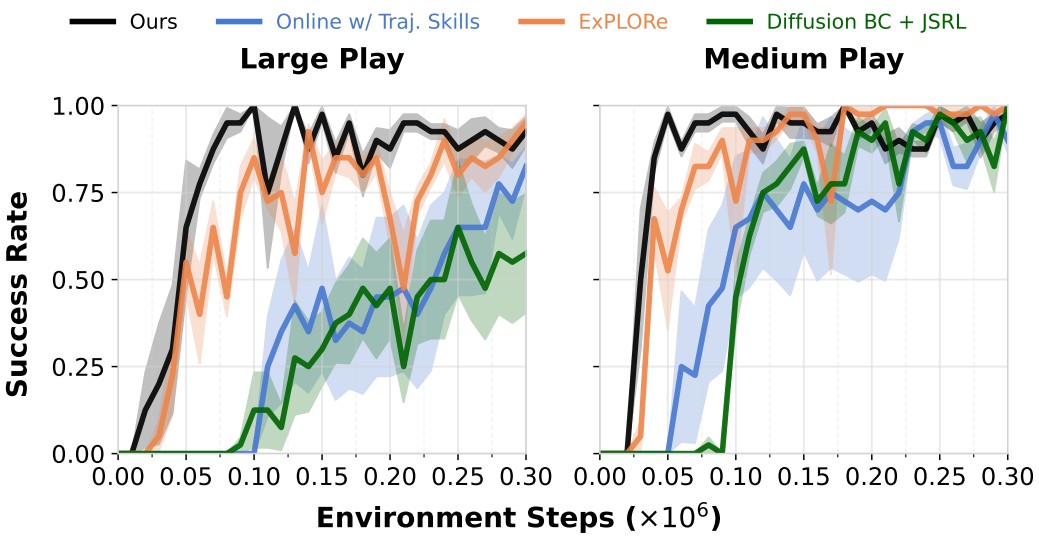

Figure 15: **Performance of our method on the play datasets. Ours** outperforms all baselines, similar to the results on the diverse datasets (Figure 11). We average over 4 seeds.

pretraining and online learning outperforms **Ours** on four of the Scene tasks, which further shows how using offline data twice is critical. Additionally, on two of the difficult Double Cube manipulation tasks, **Ours** is the only method to achieve nonzero reward. Non-skill based methods **ExPLORe** and **Diffusion BC + JSRL** performs reasonably well on the easier Cube Single and Scene task 1, but struggle to achieve significant return on the more difficult tasks, which shows how extracting structure from skills is critical for solving challenging tasks.

## J.3 ANTSOCCER

As shown in Figure 18, **Ours** matches or outperforms all baselines on AntSoccer Medium, and achieves higher final performance than all baselines on AntSoccer Arena. We also see that on both tasks, **Ours** and **HILP w/ Offline Data** outperform **Online w/ Traj. Skills** and **Online w/ HILP Skills**, which demonstrates the importance of using offline data twice to accelerate online exploration.

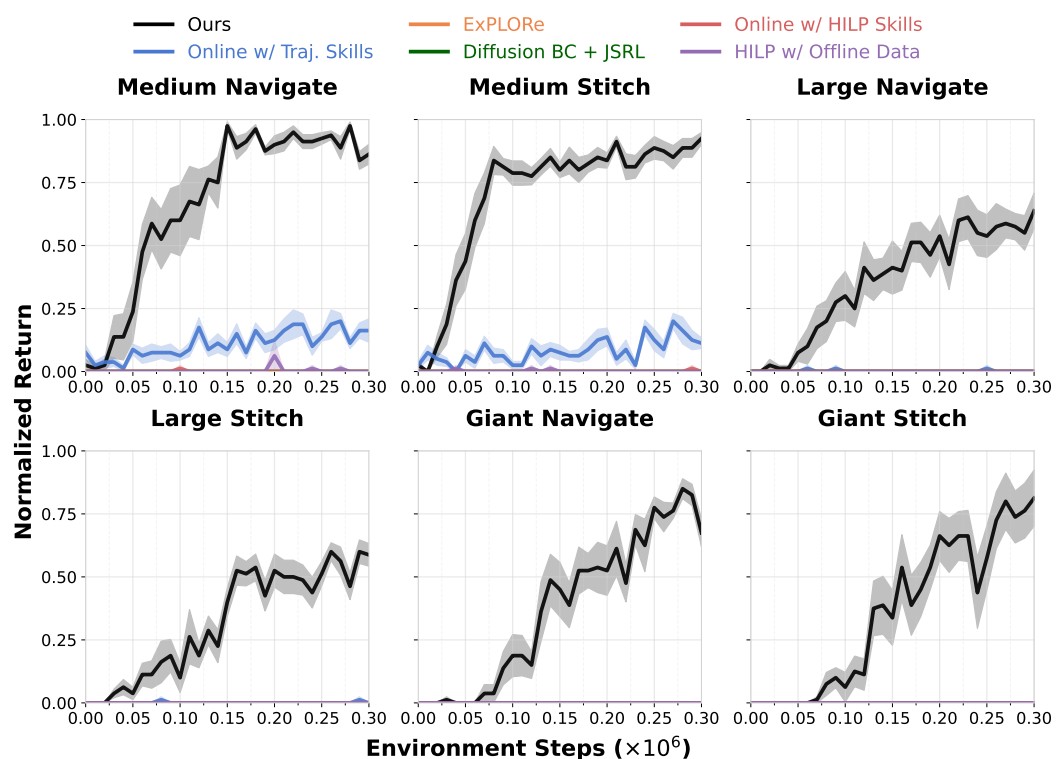

Figure 16: **Normalized return on six HumanoidMaze tasks.** Our method is the only method that solves the task with more than 50% success rate. All baselines either completely fail or only achieve less than 20% success rate on easier mazes (Online w/ Traj. Skill on `medium-navigate` and `medium-stitch`). We average over 8 seeds.

## K    SENSITIVITY TO OFFLINE DATA QUALITIES

To provide more insights on how different offline data qualities affect the performance of our method, we perform additional analysis on the `AntMaze-Large` environment.

### K.1    EXPERT DATA TO RANDOM EXPLORATORY DATA

In Figure 19, we consider four additional offline datasets for the `antmaze-large` task with decreasing dataset quality:

1. **Expert**: collected by a non-noisy expert policy that we train ourselves.
2. **Navigate**: collected by a noisy expert policy that randomly navigates the maze (from OG-Bench (Park et al., 2024a)).
3. **Stitch**: collected by the same noisy expert policy but with much shorter trajectory length (from OGBench (Park et al., 2024a))
4. **Explore**: collected by moving the ant in random directions, where the direction is re-sampled every 10 environment steps. A large amount of action noise is also added (from OGBench (Park et al., 2024a)).

As expected, the baseline ExPLORe shows a gradual performance degradation from `Expert` to `Navigate`, to `Stitch`, and to `Explore`. All skill-based methods (including our method) fail completely on `Explore`. This is to be expected because the `Explore` dataset contains too much noise and the skills extracted from the dataset are likely very poor and meaningless. The high-level policy then would have trouble composing these bad skills to perform well in the environment. On `Navigate` and `Stitch`, our method outperforms other baselines, especially on the more challenging `Stitch` dataset where it is essential to stitch shorter trajectory segments together. On `Expert`,

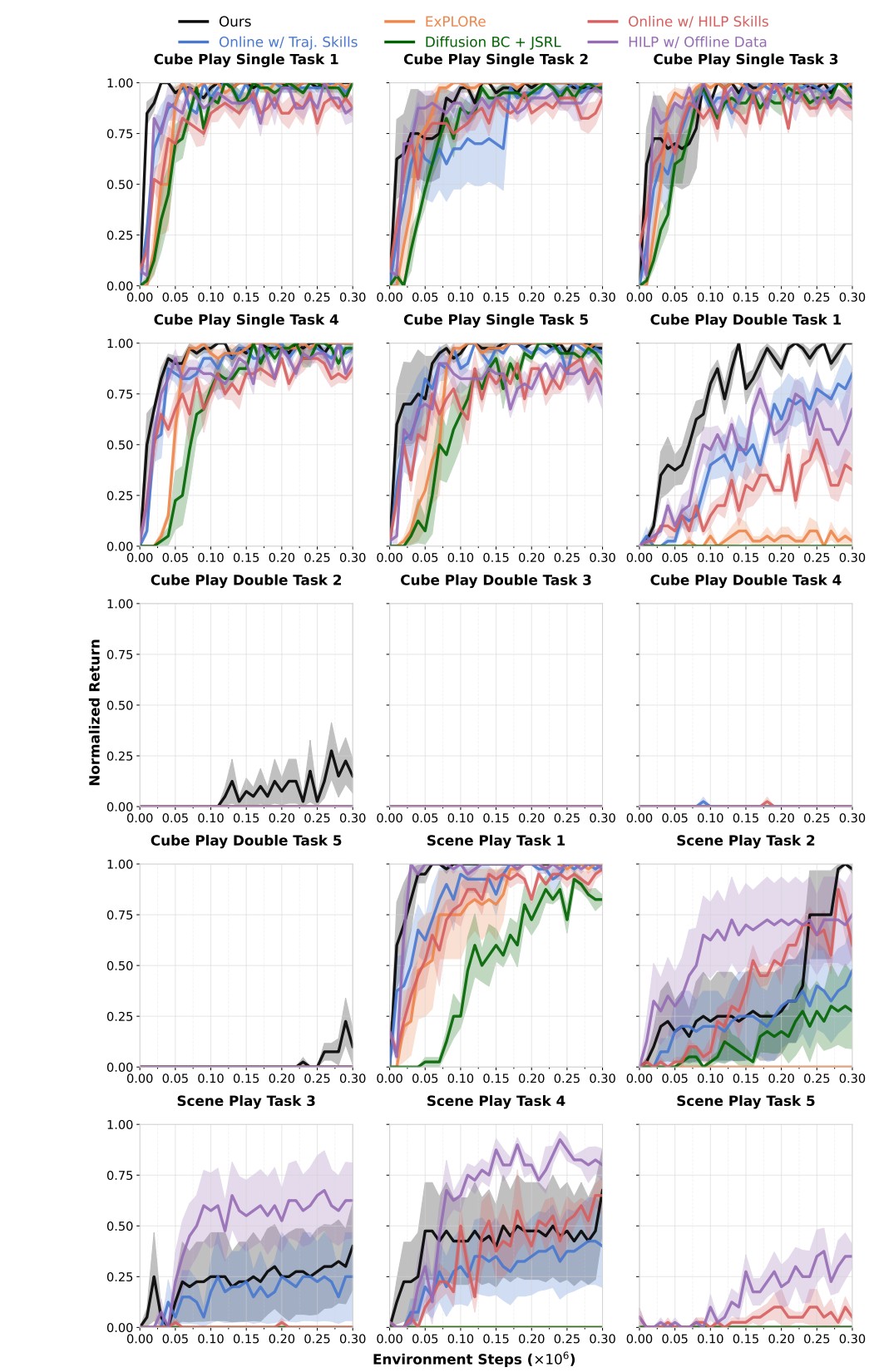

Figure 17: **Normalized return on individual tasks in Cube and Scene domains.** Cube has 10 tasks in total (5 on Cube-Single and 5 on Cube-Double). Scene has 5 tasks in total. We average over 4 seeds.

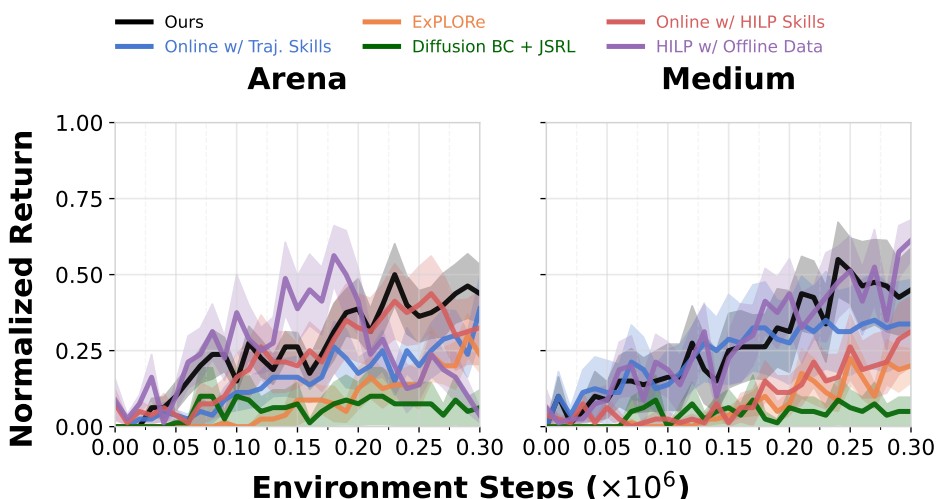

Figure 18: **Normalized return on individual tasks in the AntSoccer domain.** We average over 4 seeds.

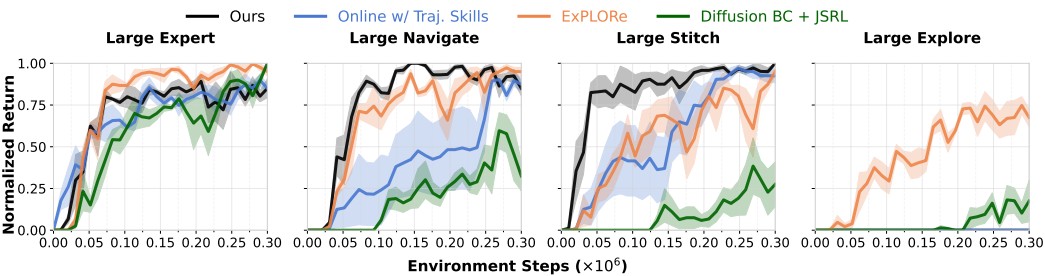

Figure 19: **Performance comparison on AntMaze-Large with different offline datasets (top-right goal).** *Expert*: collected by a non-noisy expert policy; *Navigate*: collected by a noisy expert policy that randomly navigates the maze from OGBench (Park et al., 2024a); *Stitch*: collected by the same noisy expert policy but with much shorter trajectory length (also from OGBench (Park et al., 2024a)); *Explore*: random exploratory trajectories collected by moving the ant in random directions re-sampled every 10 environment steps, with large action noise added (also from OGBench (Park et al., 2024a)).

all methods perform similarly with ExPLORe doing slightly better. We hypothesize that this is because with the expert data, online learning does not require as much exploration, and skill-based methods are mostly beneficial when there is a need for structured exploratory behaviors. Since the expert dataset has a very different distribution (much narrower) than the others, we performed a sweep over the KL coefficient in VAE training over {0.01, 0.05, 0.1, 0.2, 0.4, 0.8}, and found that 0.2 performed best, so we used these skills for both **Ours** and **Online w/ Traj. Skills** on this dataset.

## K.2 DATA CORRUPTIONS

In this section, we study how robust our method is against offline dataset corruptions. We perform an ablation study on the AntMaze domain on the large maze layout with two types of data corruption applied to the offline data:

1. *Insufficient Coverage*: All the transitions close to the goal (within a circle with a radius of 5) are removed.

2. *5% Data*: We subsample the dataset where only 5% of the trajectories are used for skill pretraining and online learning.

We report the performance on both settings in Figure 20. For the *Insufficient Coverage* setting, our method learns somewhat slower than the full data setting, but can still reach the same asymptotic performance, and outperforms or matches all baselines in the same data regime throughout the training process. For the *5% Data* setting, our method also reaches the same asymptotic performance as in the full data regime, and outperforms or matches all baselines throughout training. The gap between **Ours** and baseline performance (in particular, **ExPLORe**) is smaller than in the full data regime, which is to be expected as we have less data to learn the prior skills, so the skills are likely not as good. Overall, among the top performing methods in the AntMaze domain, our method is the most robust, consistently outperforming the other baselines that either do not use pre-trained skills (**ExPLORe**) or do not use the offline data during online learning (**Online w/ Trajectory Skills**) in these data corruption settings.

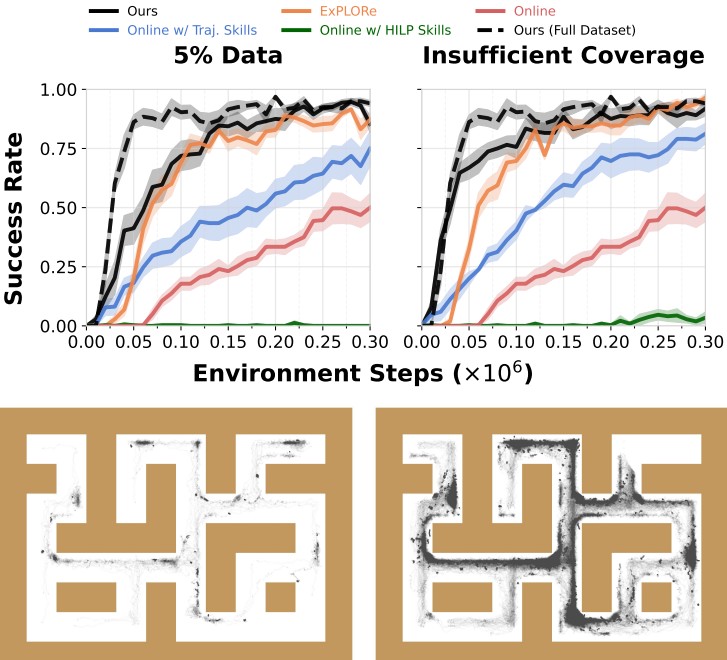

Figure 20: **Data corruption ablation on state-based `antmaze-large`.** *Top*: The success rate of different methods on these data corruption settings. *Bottom*: Visualization of the data distribution for each corruption setting. We experiment with two data corruption settings. Our method performs worse than the full data setting but still consistently outperforms all baselines.

