# OpenReview forum: "Leveraging Skills from Unlabeled Prior Data for Efficient Online Exploration"
_ICLR.cc/2025/Conference — Submitted to ICLR 2025_

### Official Review · Reviewer_t68W · 2024-10-19

**Soundness:** 2
**Presentation:** 3
**Contribution:** 2
**Rating:** 5
**Confidence:** 4

**Summary:**

This paper proposes a two-phase framework, SUPE, which leverages data in two stages: first, extracting low-level skills during the offline pre-training phase, and then using these skills and unlabeled data in the online phase to train a high-level strategy for more efficient exploration. Building on prior works like SPiRL [1] and ExPLORe [2], the key contribution of this paper is to integrate unlabeled data with online data to accelerate exploration and training in off-policy reinforcement learning (RL) methods. In the offline pre-training stage, the authors train a set of low-level skills, while in the online phase, they develop a high-level policy by utilizing both online data and relabeled offline data. To assess the method’s effectiveness, the authors compare SUPE with several baselines using benchmarks such as D4RL, and also discuss its limitations and potential directions for future research.

[1] Pertsch, Karl, Youngwoon Lee, and Joseph Lim. "Accelerating reinforcement learning with learned skill priors." Conference on robot learning. PMLR, 2021.

[2] Li, Qiyang, et al. "Accelerating exploration with unlabeled prior data." Advances in Neural Information Processing Systems 36 (2024).

**Strengths:**

* The paper is highly detailed, well-written and provides detailed motivation. The complete code is also provided.
* The authors conduct numerous experiments to thoroughly validate their method and address in detail several key issues that I am particularly concerned about, including its scalability, robustness.

**Weaknesses:**

* The overall novelty of this work is somewhat limited, as it builds heavily on existing methods and concepts (mentioned in summary).
* Although numerous experiments are conducted, the selected tasks are relatively monotonous and simplistic. The experiments test only two types of tasks: AntMaze and Kitchen.

**Questions:**

* See weakness above.
* Given the similarities between SPiRL [1] and this work, apart from the online reinforcement learning stage, why isn’t SPiRL used as a baseline for comparison (despite the numerous experiments conducted) ?
* In the pre-training stage, it would also be valuable to discuss whether trajectory segment length $H$ significantly impacts the method's performance.
* I am curious whether using expert data would result in better low-level skills during the pre-training stage.

[1] Pertsch, Karl, Youngwoon Lee, and Joseph Lim. "Accelerating reinforcement learning with learned skill priors." Conference on robot learning. PMLR, 2021.

---

> ### Author Response · Authors · 2024-11-21
> **Author Response (1/3)**
>
> Thanks for the detailed feedback and insightful comments! For your concern on the novelty of our method, we would like to highlight that important, careful design decisions in our current method enable significant performance gains over baselines whereas the naive combination of prior works falls short. For your concern on the current domains being too monotonous and simplistic, we additionally evaluated our method on four additional domains that are much more challenging and diverse than the three domains in our initial submission. We showed that our method exhibits similar performance gains on most domains. In the HumanoidMaze domain, our method is the only method that can solve all of the tasks.
>
> **Novelty.**
>
> We would like to emphasize several key design decisions in our current method that are different from prior methods, which contribute to the performance gains over baselines.
>
> First of all, all prior work on online learning with skills extracted from offline data simply discards the offline data when learning the high-level policy (e.g., Pertsch et al. (2021), Ajay et al. (2020)). In our experiments, the baseline “Online w/ Traj. Skill” does exactly this (learning trajectory skills from offline data, then learning a high level policy purely from online samples), and is consistently worse than our method that utilizes offline data during online learning (especially on more challenging tasks like the Large and Ultra AntMaze environments in Figure 8 and on all HumanoidMaze tasks in Figure 13).
>
> In addition, our method is not a naive combination of SPiRL (Pertsch et al. (2021)) and ExPLORe (Li et al. (2024)). Pertsch et al. (2021) use a KL constraint between the high-level policy and a state-dependent prior obtained from offline pretraining. We found this design can actually hurt the online performance. We show that a simpler design without the KL constraint works much better. In Figure 6, we compare (Ours (KL)) with our method (Ours) and demonstrate that the final performance and the sample efficiency of the naive combination is much worse. As described in Appendix E, we borrow the policy parameterization from Haarnoja et al. (2018) and adopt a tanh policy parameterization with entropy regularization on the squashed space. Such design ensures that the online high-level policy is not explicitly constrained to a pre-trained prior, allowing the online policy to learn skill distributions that are more suitable for the task online. It also allows the addition of entropy regularization to the high level policy, which helps exploration.
>
> These careful designs are what make our method extremely stable, sample efficient, and scalable to more complex tasks.

---

> ### Author Response · Authors · 2024-11-21
> **Author Response (2/3)**
>
> **Tasks are simplistic and monotonous.**
>
> We show additional results on four new domains (23 new tasks in total!) with two locomotion domains and two manipulation domains in Figure 3. Each of these domains contains 2-5 tasks. We selected these environments from OGBench [1], since they provided challenging, long-horizon tasks which also require exploration to solve, making them well-suited for testing the use of offline data to accelerate online exploration. The two manipulation domains (Cube and Scene) contain different tasks that are long-horizon and require composition of multiple skills by design. For example, one of the tasks in Scene requires the robotic arm to 1) press a button to unlock a drawer, 2) open the unlocked drawer, and 3) pick an object and place the object in the drawer, and 4) close the drawer. The two additional locomotion domains are HumanoidMaze and AntSoccer. HumanoidMaze is more difficult than AntMaze, since it involves controlling a 21-DoF humanoid agent. The tasks have a significantly longer time horizon, with the giant maze requiring up to 4000 environment steps. AntSoccer is also much harder than AntMaze, since the ant needs to navigate to a soccer ball and then dribble it to the goal location.
>
> In the challenging HumanoidMaze domain, our method is often the only method that achieves non-zero success rate on the four most difficult mazes. On manipulation tasks, our method consistently outperforms all prior methods on all domains with the only exception on Scene where one of the baselines (Offline w/ HILP) performs better. It is worth noting that Offline w/ HILP is a baseline that we introduced to also leverage offline data twice, both during offline and online learning with the only difference being that the unsupervised skill pre-training algorithm is HILP (instead of using trajectory VAE). This further demonstrates that the principle of leveraging offline data for both skill pre-training and online learning is effective. The effectiveness of our method across seven domains further highlights the importance of a careful combination of skill pre-training and effective online learning that utilizes the offline data.
>
> We would be happy to add additional experiments in the final version if there are specific benchmark tasks that the reviewer could suggest that they believe to be a better test for the method.
>
> [1] Park, Seohong, et al. "OGBench: Benchmarking Offline Goal-Conditioned RL." arXiv preprint arXiv:2410.20092 (2024).
>
> **Why isn’t SPiRL used as a baseline for comparison?**
>
> We already compare our method to an improved version of SPiRL. Our baseline “Online w/ Traj. Skill” in Figure 3 is an implementation of SPiRL but with additional improvements such as the addition of RND reward bonus, as well as replacing the KL constraint on the prior with entropy regularization in SAC. These improvements are crucial for online learning to work well in the challenging domains that we experiment with. This can be seen in our ablations for each of these improvements on AntMaze Large (Figure 6: KL, Figure 4, left: RND). We have made additional clarifications in our paper to reflect this.
>
> **How does trajectory segment length affect performance?**
>
> We included a set of new sensitivity analysis results in our paper on the AntMaze-Large (Top-right goal). Figure 4, right shows how the performance of our method changes as we increase and decrease the trajectory segment length (skill horizon length), H. When we decreased the length to 2, we found that our method can actually achieve an even higher final performance at the cost of slower initial learning. This is likely due to the fact that having shorter skills in the beginning makes exploration less structured, slowing down learning. Shorter skills allow the high-level policy to stitch them more optimally, improving the final performance. When we increased the skill horizon length to 8, we found that our method can still solve the task, but much more slowly. We used a constant skill length of 4 for all our experiments and we found it to work well across all the domains we tested.

---

> ### Author Response · Authors · 2024-11-21
> **Author Response (3/3)**
>
> **Can expert data lead to better skills?**
>
> We conducted an additional set of experiments that use offline data with different qualities and reported the results in Figure 16. Among these datasets, the first one is an expert dataset collected by rolling out an expert policy. We find that our method does not have advantage over non-skill-based methods (e.g., ExPLORe) and even learns worse than the setting where a more diverse dataset is provided. We hypothesize that it is because completely expert data is very narrow and can lead to a very narrow set of pre-trained skills that may not behave well in states that are out of the distribution of the offline data. This in turn can harm online learning as the high-level policy can have trouble finding low-level skills that can correct the agent from these out-of-distribution states.
>
> **Hope these address all of your questions and concerns. Thank you again for your time to review and if you have any remaining questions or concerns, please let us know!**

---

> > ### Comment · Reviewer_t68W · 2024-11-22
> >
> > Thank you for your response and the extensive additional experiments you provided. I apologize for overlooking some of the experiments detailed in the paper. Your response has addressed part of my concerns. However, considering prior work, I still think the contribution of this paper somewhat limited. I will raise the score appropriately to align with the quality of your work.

---

> > > ### Author Response · Authors · 2024-11-28
> > >
> > > Thanks again for your time to review and thanks for increasing the score. We have included some new analysis experiments (Figure 5) that provide more insights to the sensitivity of the hyperparameters in our method. Hope these experiments could further strengthen our paper. **If you have specific concerns or additional experiments that you would like to see that could improve our paper, please let us know!**

---

> > > > ### Author Response · Authors · 2024-11-30
> > > >
> > > > Thanks again for your time in reviewing. Sorry for sending multiple messages, but would you mind expanding in more detail why you believe the contribution of this paper is limited?
> > > >
> > > > If novelty is still the main concern, we would like to emphasize that **none of the prior works have effectively used offline data twice for both skill pre-training and online learning**. Our novelty lies in the observation that offline data can be simultaneously used for both skill pre-training and as additional off-policy data (which none of the prior works have observed), as well as the careful design choices that lead to strong performance in practice. ExPLORe only uses the offline data during online learning and does not perform pre-training, and prior skill-based methods did not even consider using offline data as additional prior data during online learning. In prior works, the usage of unlabeled offline data has been completely disjoint: either fully offline for skill pre-training or fully online as additional data for online learning. We show that precisely this combination is what differentiates our method in terms of performance. For example, in the HumanoidMaze environment, all of the prior methods completely failed on the more difficult Large and Giant mazes with near 0 success rate throughout training, but our method, with this combination, solves all of these tasks with high success rate (from 55\% to 80\%).
> > > >
> > > > In addition, we provided the necessary design details (e.g., removing the KL) that enables our method to achieve such sample efficiency, and we showed empirically that this design is beneficial. While our paper does not focus on these design details, we believe it is still a very valuable contribution to the community because **it, along with using offline data for both pre-training and as additional data for online RL, enables online RL to operate at a level of sample efficiency significantly better than previous state-of-the-art.**
> > > >
> > > > **We hope this can help address your concerns, and if there are any new experiments or specific questions that could help us improve the paper further, please let us know! Thanks again!**

---

### Official Review · Reviewer_CDzd · 2024-10-22

**Soundness:** 2
**Presentation:** 3
**Contribution:** 2
**Rating:** 6
**Confidence:** 3

**Summary:**

This paper presents a pre-training method for reinforcement learning (RL) that can train on data sets that do not contain reward labels, i.e., the data sets are unlabeled.
The problem setting resembles offline-to-online RL, except that there are no rewards in the data set.
In the pre-training stage, the authors propose to learn a set of skills from this unlabeled offline data.
Then, in the online fine-tuning state, the authors learn a high-level policy that selects which skill to use in a given state.
They utilize the unlabeled offline data during fine-tuning by learning an optimistic reward model and using it to add optimistic reward labels to the offline data.
They evaluate their method in the D4RL AntMaze and Kitchen benchmarks as well as the D4RL Visual AntMaze.

**Strengths:**

Overall, I found the paper easy to follow and I think it is addressing an important problem -- pre-training in RL -- which is of interest to the community.

The results demonstrate that learning skills from offline data is a promising approach to leverage reward-free offline data.
I think this is an interesting result.
I also like the idea of labelling the offline data using a learned reward function.

**Weaknesses:**

The authors consider the setting of having access to offline data but no reward labels.
Whilst I see the value in this problem setting, it is not clear if practitioners should opt for this method over standard offline-to-online RL methods when
their data sets contain reward labels.
Whilst I appreciate this is out-of-scope, ideally methods would leverage data sets both with and without reward labels.
It would be insightful if the authors could compare to offline-to-online RL methods which do leverage reward labels.
Whilst I do not expect their method to outperform these methods, I think it is an important baseline that we can gain insights from.

In my experience, optimistic-based exploration methods are very susceptible to the $\alpha$ parameter.
How was this set in practice?
Did it require a grid search to find the best value in each environment?
Please can you provide details on any hyperparameter tuning process, including the range of values tested and how sensitivity varied across environments?
This information would be valuable for reproducibility and understanding the robustness of the method.

Is there a reason the authors only considered the diverse data set for the AntMaze experiments?
Does this method require a diverse offline data set collected by an unsupervised RL method,
or can it leverage narrow offline data distributions? For example, data from solving a different task?
How does the method perform when using the AntMaze "play" data set instead of the "diverse" data set?
Even if the method performs poorly, I think it would be valuable to include these results.

I am not sure what to take from the coverage results.
I can understand why we care about coverage in unsupervised RL where our sole purpose is to explore.
However, during online training our goal is to balance exploration vs exploitation.
Please can the authors provide a clearer justification for why coverage is an important metric in this context, or include additional plots that more directly show the relationship between exploration and task performance, such as the normalized return vs coverage?

In Table 1, what do the bold numbers represent? The authors should state what statistical test was used for the bolding or at least expla8in what the bolding represents.

## Minor comments and corrections
- Line 42 - "can broken" should be "can be broken"
- Line 117 - "of an offline data" should be "of offline data"
- Line 200 - the term "latent code" is misleading. This suggests the trajectory encoder learns to map trajectories to discrete codes from a codebook and I don't think this is the case. The authors should change it to something like "latent skill".
- Line 279 - Should "Three AntMaze layouts with four different goal location configuration each." be "Three AntMaze layouts with four configurable goal locations each."
- Line 411-414 - It would make more sense for this paragraph to be at the start of Section 5.4.
- Line 407 - "Kitchen the domain" should be "Kitchen domain"
- Line 408 - "more challenging the kitchen-mixed" should be "more challenging kitchen-mixed"
- Figures - The authors have stated that the shaded area indicates the standard error. They also need to state what that solid line indicates. Is it the mean, median, etc?
- Figures - I found the figures very hard to read. I would suggest the authors colour the text "HILP w/ Offline Data", "Ours", "Online w/ HILP Skills", etc, to match the colours of the lines in the plots. This would make the text/figures much easier to read.

**Questions:**

- How does your method compare to using offline-to-online RL methods which have access to reward labels?
- How was the $\alpha$ hyperparameter set?
- Why did you not compare to other types of offline data sets?
- What should I take from the coverage results?
- In Table 1, what do the bold numbers represent?

---

> ### Author Response · Authors · 2024-11-21
> **Author Response (1/2)**
>
> Thanks for the detailed feedback and insightful comments! We addressed your concern on sensitivity of the hyperparameter by adding a new section in our experimental results that is dedicated to sensitivity analysis of various hyperparameters and how they were chosen (RND coefficient, skill horizon length). We also provided additional results on offline data with different qualities (expert/exploratory, noisy/non-noisy) with more insights on when our method is expected to work or not work depending on the dataset. Finally, we demonstrated that our method, when given access to ground truth offline reward, can outperform the state-of-the-art offline-to-online RL methods, showcasing the effectiveness of our method at leveraging structured exploration with skills.
>
>
> **How was the RND coefficient $\alpha$ set?**
>
>
> We did not find the performance of our approach to be sensitive to the RND coefficient. We ran a small hyperparameter tuning sweep on AntMaze-Large (top-right goal) and displayed the results in Figure 4, left. The performance remains almost the same for $\alpha \in \{2, 4, 8, 16\}$. We picked one of these values ($\alpha=8$) and used the SAME value for all other tasks. All the non-skill-based methods use a RND coefficient that is $4\times$ smaller to keep the reward scale consistent with the skill-based methods. This is because the transitions in skill-based methods have a reward value that is equal to the sum of the rewards in 4 low-level transitions (H=4 is the skill horizon length).
>
>
> **Additional datasets for AntMaze**
>
>
> We include the results for the play datasets for the medium maze layout and the large maze layout (Figure 12). The results are consistent with our results on the diverse datasets.
>
>
> In addition, we experimented on a narrow, expert dataset and three other datasets used in an offline goal-conditioned RL benchmark (OGBench [1]), shown in Figure 16. We picked datasets from this benchmark as it features a more diverse set of offline data distributions and hoped that it would provide more insights on when our method works or fails. As a summary, we find that our method does not need a dataset that is collected by unsupervised RL agents. Completely exploratory dataset can actually break our method due to the lack of behavioral structure that can be extracted as skills. Our method excels at learning from datasets that contain segments of meaningful (e.g., navigating around the maze) behaviors. We discuss the results in detail below.
>
>
> The four datasets we consider are ordered in decreasing difficulty:
> - Expert: collected by a non-noisy expert policy that we train ourselves.
> - Navigate: collected by a noisy expert policy that randomly navigates the maze (from OGBench).
> - Stitch: collected by the same noisy expert policy but with much shorter trajectory length (from OGBench)
> - Explore: collected by moving the ant in random directions, where the direction is re-sampled every 10 environment steps. A large amount of action noise is also added (from OGBench).
>
>
> As expected, the baseline ExPLORe shows a gradual performance degradation from Expert to Navigate, to Stitch, and to Explore. All skill-based methods (including our method) fail completely on Explore. This is to be expected because the Explore dataset contains too much noise and the skills extracted from the dataset are likely very poor and meaningless. The high-level policy then would have trouble composing these bad skills to perform well in the environment. On Navigate and Stitch, our method outperforms other baselines, especially on the more challenging Stitch dataset where it is essential to stitch shorter trajectory segments together. On Expert, all methods perform similarly with ExPLORe doing slightly better. We hypothesize that this is because with the expert data, online learning does not require as much exploration, and skill-based methods are mostly beneficial when there is a need for structured exploratory behaviors.
>
>
> To further test our method’s ability to handle different offline data quality, in our original submission, we included ablation studies where the offline data is corrupted. We tested a dataset without transitions near the goal location (Insufficient Coverage), and a low-data setting where 95% of the trajectories are removed from the dataset (5% Data). While we see performance degradation from such data erasure, our method is still the most robust, consistently outperforming the baselines (Figure 17).
>
>
> We hope that these additional experiments provide more insights on when we expect our skill-based method to work or not work.
>
>
> [1] Park, Seohong, et al. "OGBench: Benchmarking Offline Goal-Conditioned RL." arXiv preprint arXiv:2410.20092 (2024).

---

> ### Author Response · Authors · 2024-11-21
> **Author Response (2/2)**
>
> **How does your method compare to using offline-to-online RL methods which have access to reward labels?**
>
>
> In Figure 7, we show a comparison between our method and two state-of-the-art offline-to-online methods (CalQL [1] and IDQL [2]) on AntMaze-large (top-right goal). We also include a version of our method (Ours (Ground Truth)) where we assume access to the ground truth reward similar to all the offline-to-online RL methods. While Ours performs slightly worse than CalQL since we do not assume access to the offline reward, Ours (Ground Truth) performs better than CalQL with a much faster initial learning thanks to structured exploration using pre-trained skills.
>
>
> [1] Nakamoto, Mitsuhiko, et al. "Cal-QL: Calibrated offline RL pre-training for efficient online fine-tuning." Advances in Neural Information Processing Systems 36 (2024).
>
>
> [2] Hansen-Estruch, Philippe, et al. "IDQL: Implicit Q-learning as an actor-critic method with diffusion policies." arXiv preprint arXiv:2304.10573 (2023)
>
>
> ## Questions
>
>
> **What should I take from the coverage results?**
>
>
> Since we DO NOT have reward labels in the dataset and the reward information is NOT given to the agent in the beginning of the online phase, on AntMaze, the agent must find the goal first before it can learn to reach it consistently. The coverage metric provides information on how quickly the agent is able to explore the maze. However, we do agree that the coverage plot adds little in addition to Table 1 (which already provides insights on how quickly each method can find the goal). We have moved it to the Appendix in the interest of space.
>
>
> **What do the bold numbers represent in Table 1?**
>
>
> The table values are bolded if their confidence intervals overlap with the confidence interval of the best method (determined by the mean value) in each row. We included additional clarification in our paper.
>
>
> **Typos and minor comments**
>
>
> Thanks for the detailed comments! We have fixed all the typos mentioned. We also moved the last paragraph of Section 5.3 to the beginning of Section 5.4. We also added a clarification in the paper about what the solid line indicates. It represents the mean value across seeds. We also improved all the figures such that the colors of the legend text match the line colors.
>
>
> **Hope these address all of your concerns. Thank you again for your time to review and if you have any remaining questions or concerns, please let us know!**

---

> > ### Comment · Reviewer_CDzd · 2024-11-27
> >
> > I thank the authors for their efforts in addressing my questions. I have read your response and promise to review the changes carefully.

---

> > ### Comment · Reviewer_CDzd · 2024-11-27
> >
> > 1. **How was the RND coefficient set?** The sensitivity analysis on $\alpha$ and horizon $h$ are a good start. However, I am not sure that evaluating on a single task is enough. I would encourage the authors to evaluate on more tasks and report aggregate metrics (e.g. IQM) with stratified confidence intervals following [Rliable](https://github.com/google-research/rliable). I am also not sure that a horizon of $h=4$ really strikes a good balance. The results in AntMaze Large suggest that horizon h=2 performs significantly better? Again, the authors should include results on more tasks.
> > 2. **Additional datasets for AntMaze** This is a very nice addition to the paper, thank you for adding it.
> > 3. **How does your method compare to using offline-to-online RL methods which have access to reward labels?** Again, this is a good start, but I do not trust the results obtained in a single task. Please add more tasks to improve the empirical study.
> > 4. **What do the bold numbers represent in Table 1?** I do not think it is OK to bold based on overlapping confidence intervals as this does not indicate statistical significance. Please perform a t-test.
> >
> > I thank the authors for their hard work in addressing my comments. Whilst I think the paper has improved, I still have concerns regarding statistical significance in reporting results, so I will maintain my score.

---

> > > ### Author Response · Authors · 2024-11-28
> > >
> > > Thanks for the additional questions and comments.
> > >
> > >
> > > We are running new requested experiments right now on all of the 7 domains that we evaluated in our paper. In our rebuttal, we added results on 23 new tasks across 4 new domains which further showcase the effectiveness of our method  (Figure 3, now using the rliable library to generate the aggregation plots). We provide a response for the questions and concerns that we can address below, and we will follow-up with an additional response upon the completion of our experiments before the rebuttal period ends.
> > >
> > >
> > > **"I am also not sure that a horizon of h=4 really strikes a good balance. The results in AntMaze Large suggest that horizon h=2 performs significantly better? Again, the authors should include results on more tasks."**
> > >
> > >
> > > As we mentioned in our analysis (Section 5.5), H=4 explores much faster in the beginning (achieving a much higher initial success rate). H=2 achieves a better final performance but learns much slower in the beginning. We do not only  look at the final performance, but also the exploration efficiency in the beginning. We are running more experiments on new environments and we will update the thread and include the new results when they are finished.
> > >
> > >
> > > **"I do not think it is OK to bold based on overlapping confidence intervals as this does not indicate statistical significance. Please perform a t-test."**
> > >
> > >
> > > We redid the bolding of the table using a t-test with p=0.05. Please see the updated table (Table 3, blue indicates newly bolded values). We also included a new summary plot that uses the aggregated IQM with 95% stratified bootstrap confidence intervals (using rliable) in Figure 4. The plot shows that our method is the most effective with statistical significance.
> > >
> > >
> > >
> > >
> > > **"I would encourage the authors to evaluate on more tasks and report aggregate metrics (e.g. IQM) with stratified confidence intervals following Rliable."**
> > >
> > >
> > > In Figure 3, we show the comparison of our method with baselines on 7 domains (in total 42 tasks, We describe these domains in Section 5.1 in short and in Appendix D with more detail). We also include an aggregated result plot (Figure 4, IQM with 95% Stratified Bootstrap CIs plotted using the RLiable library) which shows that our method has the best sample efficiency and best final performance. In particular, on the HumanMaze domain, our method is the only method that solves all tasks whereas all other methods fail almost completely (see Figure 13 for the individual tasks in the Humanoid domain).
> > >
> > >
> > > **Hope these address most of your concerns. For the remaining concerns on the sensitivity analysis and the ground-truth reward experiments, we will post a follow-up response once the experiments are done. Thank you again for your time to review and if you have any other remaining questions or concerns, please let us know!**

---

> > > > ### Author Response · Authors · 2024-11-28
> > > > **Follow-up on the new experiments**
> > > >
> > > > Thanks for your patience! We would like to give an update on the latest experimental results for our sensitivity analysis (Section 5.5 - Figure 5: aggregated, Appendix H, Figure 9 and 10: individual domains) and ground-truth reward experiments (Appendix G, Figure 8).
> > > >
> > > > **Sensitivity Analysis**
> > > >
> > > > Our sensitivity analysis experiments were done and we have reported our results (aggregated metrics with IQM and stratified confidence interval following rliable over **7 tasks**) in Figure 5. Our sensitivity analysis shows that while having non-zero RND coefficient value is important, our method is not very sensitive to the RND coefficient value. For the skill horizon length, we find that H=4 is the best (among other values H=2 and H=8) when aggregating over tasks. *Ours with H=2* only did well on the AntMaze task in terms of the final performance. On all other tasks, *Ours with H=2* performs worse than *Ours with H=4* throughout training.
> > > >
> > > > **Comparison with Offline-to-Online Methods with Ground Truth Reward**
> > > >
> > > > Our ground-truth reward experiments were not fully completed by the PDF update deadline, but we have included the results we have so far in Figure 8 in the Appendix. We now have seven tasks (4 AntMaze tasks and 3 Kitchen tasks). The curves for both IDQ and CalQL are taken from the paper. IDQL only includes training curves for AntMaze Large. CalQL includes curves for all seven tasks, but for Kitchen they only have results for update-to-data ratio, UTD=1, while our method is UTD=20. Given that there was less than a day from the reviewer response to the PDF deadline, we did not have sufficient compute to reproduce IDQL and CalQL curves on all seven environments with the same UTD as our method. We are working these experiments right now with UTD=20 for a more fair comparison and we will provide a follow-up update once the results are out.
> > > >
> > > > Across all four AntMaze tasks and two of the three Kitchen tasks, our method is able to outperform SOTA offline-to-online methods Cal-QL and RLPD. On the AntMaze-Large-Diverse and AntMaze-Large-Play tasks where we have results for IDQL, our method is better than IDQL.
> > > >
> > > >  **Hope this addresses most of your concerns. We will post a follow-up update once the experiments for the offline-to-online comparison are fully completed. We hope that the current offline-to-online experiments still provide enough evidence that our method is effective, even under a setting that our method is not designed for. Thank you again for your time to review and if you have any other remaining questions or concerns, please let us know!**

---

> > > > > ### Comment · Reviewer_CDzd · 2024-12-03
> > > > >
> > > > > Thank you for addressing my concerns, especially those regarding statistical significance. In particular, thank you for updating the table so that bolding represents a t-test and for reporting aggregate metrics. I believe this improves the empirical study and the conclusions we can draw from it. I will raise my score accordingly.
> > > > >
> > > > > One minor point about Figure 4. For the "All Mazes" plot the number of environment steps in each environment should probably be normalized before aggregating them.

---

### Official Review · Reviewer_DC3r · 2024-11-01

**Soundness:** 2
**Presentation:** 3
**Contribution:** 2
**Rating:** 5
**Confidence:** 4

**Summary:**

This paper introduces SUPE, a method that leverages unsupervised learning to extract skills from unlabeled prior data, subsequently using hierarchical methods to explore more efficiently. These unlabeled data can also contribute to high-level policy training. Experimental results show that SUPE outperforms previous methods on the D4RL benchmark.

**Strengths:**

- The approach of extracting latent “skills” from unlabeled data and employing hierarchical methods significantly enhances exploration.
- The approach of utilizing prior data twice ensures better use of the available data.
- The paper is well-structured and easy to follow.
- Extensive results demonstrate that this method outperforms previous approaches.

**Weaknesses:**

- The concept of using a VAE to extract latent codes and employing a high-level policy for online exploration is not novel, and it shows limited progress compared to previous work [1].
- The ablation study lacks depth. I am interested in understanding the contribution of “reusing prior data twice” to the final performance. Additionally, I’d like clarification on the design choice for the latent variable $z$ in skill discovery: how do you ensure this latent $z$  is sufficient for effective skill discovery in the dataset? Is employing trajectory-segment VAEs truly necessary for efficient exploration?


[1] Qiyang Li, Jason Zhang, Dibya Ghosh, Amy Zhang, and Sergey Levine. Accelerating exploration with unlabeled prior data. Advances in Neural Information Processing Systems, 36, 2024.

**Questions:**

Please refer to the weakness part. I may consider increasing the score if my questions are addressed.

---

> ### Author Response · Authors · 2024-11-21
> **Author Response (1/2)**
>
> Thanks for the detailed feedback and insightful comments! For your concern on the novelty of our method, we would like to highlight that important, careful design decisions in our current method enable significant performance gains over baselines whereas the naive combination of prior works falls short. We additionally evaluated our method on four additional domains and our method exhibits similar performance gains. We provided clarification on how each of our baselines use prior data and showcase the effectiveness of the principle of “reusing prior data twice” on the original domains and the new four domains that we added in this rebuttal.
>
> **Novelty**
>
> We would like to emphasize several key design decisions in our current method that are different from prior methods, which contribute to the performance gains over baselines.
>
> First of all, all prior work on online learning with skills extracted from offline data simply discards the offline data when learning the high-level policy (e.g., Pertsch et al. (2021), Ajay et al. (2020)). In our experiments, the baseline “Online w/ Traj. Skill” does exactly this (learning trajectory skills from offline data, then learning a high level policy purely from online samples), and is consistently worse than our method that utilizes offline data during online learning (especially on more challenging tasks like the Large and Ultra AntMaze environments in Figure 11 and on all HumanoidMaze tasks in Figure 16).
>
> In addition, our method is not a naive combination of SPiRL (Pertsch et al. (2021)) and ExPLORe (Li et al. (2024)). Pertsch et al. (2021) use a KL constraint between the high-level policy and a state-dependent prior obtained from offline pretraining. We found this design can actually hurt the online performance. We show that a simpler design without the KL constraint works much better. In Figure 7, we compare (Ours (KL)) with our method (Ours) and demonstrate that the final performance and the sample efficiency of the naive combination is much worse. As described in Appendix E, we borrow the policy parameterization from Haarnoja et al. (2018) and adopt a tanh policy parameterization with entropy regularization on the squashed space. Such design ensures that the online high-level policy is not explicitly constrained to a pre-trained prior, allowing the online policy to learn skill distributions that are more suitable for the task online. It also allows the addition of entropy regularization to the high-level policy, which helps exploration.
>
> These careful designs are what make our method extremely stable, sample efficient, and scalable to more complex tasks. We show additional results on four new domains with two locomotion domains and two manipulation domains in Figure 3. We selected these environments from OGBench [1], since they provided challenging, long-horizon tasks which also require exploration to solve, making them well-suited for testing the use of offline data to accelerate online exploration. In the challenging HumanoidMaze domain, our method is often the only method that achieves non-zero success rate on the four most difficult mazes. On manipulation tasks, our method consistently outperforms all prior methods on all domains with the only exception on Scene where one of the baselines (Offline w/ HILP) performs better. It is worth noting that Offline w/ HILP is a novel baseline that we introduced to also leverage offline data twice, both during offline and online learning with the only difference being that the unsupervised skill pre-training algorithm is HILP (instead of using trajectory VAE). This further demonstrates that the principle of leveraging offline data for both skill pre-training and online learning is effective. The effectiveness of our method across seven domains further highlights the importance of a careful combination of skill pre-training and effective online learning that utilizes the offline data.
>
> [1] Park, Seohong, et al. "OGBench: Benchmarking Offline Goal-Conditioned RL." arXiv preprint arXiv:2410.20092 (2024).

---

> ### Author Response · Authors · 2024-11-21
> **Author Response (2/2)**
>
> **Importance of Reusing Prior Data Twice**
>
> We would like to clarify that the baseline Online w/ Traj. Skill only uses the offline data for unsupervised skill pretraining and the baseline ExPLORe only uses the offline data during online learning and does not have a skill pre-training phase. In Figure 3, we show that none of these baseline methods that only use the offline data once can achieve good performance. Our method consistently beats these two baselines, demonstrating that reusing the prior data twice is crucial.
>
> In addition, one of the baselines that we introduce in this work, Online w/ HILP Skills, also only uses the offline data for unsupervised skill pretraining. We apply the same principle of reusing the prior data twice to this baseline which leads to the HILP w/ Offline Data baseline. Across domains, HILP w/ Offline Data consistently outperforms Online w/ HILP Skills, further highlighting the benefits of reusing the prior data during online learning.
>
> **Trajectory-Segment VAE**
>
> Trajectory-segment VAE is a common design choice for extracting a latent space of skill policies offline adopted by a range of prior works and has shown effectiveness in accelerating RL [1-2]. While such a design is certainly not the only approach that can extract useful skills from offline data, it is the simplest formulation that we found to be effective. In addition, the trajectory encoder in the VAE allows to conveniently transform the offline data into high-level off-policy data such that they can be readily used by the actor-critic RL agent online as additional off-policy data, allowing us to use the offline data twice. In addition, the idea of “reusing the prior data twice” can be applied to potentially other unsupervised skill pre-training algorithms. In our work, we present one alternative where we use HILP [3], a recently proposed offline unsupervised skill pre-raining method and implement two baselines. The first baseline “Online w/ HILP Skill” is the naive version that does not use the prior data twice (the online learning does not use the offline data as additional off-policy data). The second baseline “HILP w/ Offline Data” is the version that does use the prior data twice and we observe that the second baseline (that uses the data twice) performs consistently better than the first baseline (that only uses the data once) across all the domains (Figure 3).
>
> [1] Pertsch, Karl, Youngwoon Lee, and Joseph Lim. "Accelerating reinforcement learning with learned skill priors." Conference on robot learning. PMLR, 2021.
>
> [2] Ajay, Anurag, et al. "Opal: Offline primitive discovery for accelerating offline reinforcement learning." arXiv preprint arXiv:2010.13611 (2020).
>
> [3] Park, Seohong, Tobias Kreiman, and Sergey Levine. "Foundation policies with hilbert representations." arXiv preprint arXiv:2402.15567 (2024).
>
>
> **Hope these address all of your questions and concerns. Thank you again for your time to review and if you have any remaining questions or concerns, please let us know!**

---

### Official Review · Reviewer_28dh · 2024-11-03

**Soundness:** 3
**Presentation:** 4
**Contribution:** 3
**Rating:** 8
**Confidence:** 2

**Summary:**

This paper presents SUPE, a method for using offline data (without rewards) in the online reinforcement learning setting. SUPE first extracts a set of low level skills using the offline data, and then optimistically labels the offline trajectories. It then uses an off policy high level update to update on a mix of offline (pseudo labeled trajectories) and online real trajectories. The paper empirically validates the new algorithm on three environments and does ablations on amounts of offline data.

**Strengths:**

- This paper makes an insightful empirical benefit for using trajectories twice for both low level skill pretraining in addition to optimistic labelling.
- The paper thoroughly evaluates the proposed method.
- The paper does a good job explaining the proposed method and it's significance.

**Weaknesses:**

- This paper could benefit from a bit deeper analysis of the contribution of the two uses of offline data. It's clear that both are necessary, but not necessarily why.

**Questions:**

- Where do the authors think their empirical benefit is coming from? Why can we use trajectories twice?
- Is the algorithm robust to different design choices?
- How important is the optimistic labelling (from Li et al.)?

---

> ### Author Response · Authors · 2024-11-21
>
> Thanks for the detailed feedback and insightful comments! To address your questions on our design choices, we provided additional sensitivity analysis on design choices in our algorithm. We also provided more discussions on the contribution of the two uses of offline data. Hope these help provide more insights on how our method works.
>
> **Is the algorithm robust to different design choices? How important is the optimistic labelling?**
>
> We include additional sensitivity analysis on the skill horizon (H) and the RND coefficient (the amount of optimism added in optimistic labeling) using the AntMaze-Large (top-right goal) task.
>
> Optimistic labeling is important for our method to successfully solve the task. In Figure 5, left, our method with no optimistic labeling ($\alpha=0$) completely fails to solve the task. However, the performance of our method is robust to the value of the RND coefficient as long as it is non-zero –  the performance remains almost the same for $\alpha \in \\{2, 8, 16\\}$.
>
> For the skill horizon (H), Figure 5, right shows how the performance of our method changes as we increase and decrease the trajectory segment length (skill horizon length), H. We find that while there is some variability across individual tasks (Appendix H, Figure 10), a skill horizon length of 4 generally performs the best. We used a constant skill length of 4 for all our experiments and we found it to work well across all the domains we tested.
>
> **Why can we use the offline data trajectories twice?**
>
> The use of offline data during offline pre-training and online learning are along two axes that complement each other. Offline pre-training leverages the short horizon behavioral structure in offline dataset whereas online learning leverages the more high-level dynamics information of the environment (e.g., how a state at the current time step $s_t$ may be transformed to a state $H$ steps in the future $s_{t+H}$ via high-level skills). The behavioral structure helps construct the skills and the high-level dynamics information helps stitching/composing these skills together at a higher-level. Since our high-level policy is an off-policy RL agent, it can consume off-policy high-level transitions directly from the offline data to help it stitch/compose the low-level skills. This can be further justified by observing that on the domains where stitching is needed less (e.g., Single Cube tasks and Kitchen-complete where the demonstration of completing the full task is directly available in the offline data), the gap between our method and the Online w/ Traj. Skills baseline (that does not use the offline data trajectories during online learning) is lower.
>
> **Hope these address all of your questions and concerns. Thank you again for your time to review and if you have any remaining questions or concerns, please let us know!**

---

> > ### Comment · Reviewer_28dh · 2024-11-28
> > **response to authors**
> >
> > Thank you for your comments and addressing my questions. I believe that authors have still not addressed the weaknesses I mentioned, in particular some motivation for the empirical benefit for both uses of offline data is still requested. However, I will maintain my score as I still believe this is a good paper.

---

### Official Review · Reviewer_eVtj · 2024-11-03

**Soundness:** 3
**Presentation:** 3
**Contribution:** 2
**Rating:** 5
**Confidence:** 4

**Summary:**

The paper proposes a hierarchical policy for leveraging unlabeled offline data for exploration. In the offline stage, low-level skills are extracted, and in the online stage, these skills are reused and a high-level policy is learned with optimistic rewards. The proposed method is tested on maze and manipulation tasks and shows good performance.

**Strengths:**

- The paper is well-written and easy to understand.
- The paper proposes a simple method for leveraging offline data and showing good performance on AntMaze, visual AntMaze, and Kitchen tasks.
- The paper conducts thorough experiments and compares a set of different methods.

**Weaknesses:**

- Dependence on offline data quality: The performance of the proposed method is influenced by the quality of the offline data and the specific features of the evaluation tasks. In particular, the approach relies on a high-level policy that is updated every
𝐻 timesteps and keeps the pre-trained skill and trajectory encoder fixed during the online phase. This limitation constrains adaptability, especially in scenarios where task distribution varies from the offline data.
- Limited discussion on Hierarchical Reinforcement Learning (HRL): Although hierarchical policy structures have been extensively explored in the HRL literature [1-8] and are closely related to the paper, the paper does not sufficiently address relevant findings from HRL research. A more comprehensive discussion of how this work could provide valuable context.
- Novelty: The paper combines elements from ExPLORe and trajectory-segment VAE to leverage offline data for exploration, but adds limited new insights beyond prior work. HRL emphasizes hierarchical structures, and the benefits of skill extraction in offline settings have already been documented. This paper simply applies existing solutions to ExPLORe.

The paper could be improved in several aspects:
- Refinement of skill extraction method: Currently, skills are extracted based on fixed-length trajectory segments, a method that may overlook important nuances in skills. A more flexible or adaptive approach could address these limitations, potentially enhancing the robustness of the extracted skills.
- Skill adaptation during the online stage: The method does not allow for online adaptation of the skill policy or trajectory encoder. Due to potential distributional shifts between the offline and online data, enabling adaptive updates to the skill set and encoder could further improve the performance.
- Training stability in HRL is often affected by interactions between high-level and low-level policies. This work could benefit from discussing how offline data might address or mitigate these stability challenges.

[1] Kulkarni, Tejas D., et al. "Hierarchical deep reinforcement learning: Integrating temporal abstraction and intrinsic motivation." Advances in neural information processing systems 29 (2016).

[2] Xie, Kevin, et al. "Latent skill planning for exploration and transfer." arXiv preprint arXiv:2011.13897 (2020).

[3] Nachum, Ofir, et al. "Data-efficient hierarchical reinforcement learning." Advances in neural information processing systems 31 (2018).

[4] Bacon, Pierre-Luc, Jean Harb, and Doina Precup. "The option-critic architecture." Proceedings of the AAAI conference on artificial intelligence. Vol. 31. No. 1. 2017.

[5] Ajay, Anurag, et al. "Opal: Offline primitive discovery for accelerating offline reinforcement learning." arXiv preprint arXiv:2010.13611 (2020).

[6] Gehring, Jonas, et al. "Hierarchical skills for efficient exploration." Advances in Neural Information Processing Systems 34 (2021): 11553-11564.

[7] Dalal, Murtaza, Deepak Pathak, and Russ R. Salakhutdinov. "Accelerating robotic reinforcement learning via parameterized action primitives." Advances in Neural Information Processing Systems 34 (2021): 21847-21859.

[8] Paraschos, Alexandros, et al. "Probabilistic movement primitives." Advances in neural information processing systems 26 (2013).

**Questions:**

See weaknesses above.

---

> ### Author Response · Authors · 2024-11-21
> **Author Response (1/3)**
>
> Thanks for the detailed feedback and insightful comments. For your concern on the novelty of our method, we would like to highlight that important, careful design decisions in our current method enable significant performance gains over baselines whereas the naive combination of prior works falls short. We additionally evaluated our method on four additional domains and our method exhibits similar performance gains. We also addressed your concern on the lack of discussion on HRL by adding a new paragraph in the related work to discuss prior works in HRL and provided justification of our choice of the skill formulation.
>
> **Novelty**
>
> We would like to emphasize several key design decisions in our current method that are different from prior methods, which contribute to performance gains over baselines.
>
> First of all, all prior work on online learning with skills extracted from offline data simply discards the offline data when learning the high-level policy (e.g., Pertsch et al. (2021), Ajay et al. (2020)). In our experiments, the baseline “Online w/ Traj. Skill” does exactly this (learning trajectory skills from offline data, then learning a high level policy purely from online samples), and is consistently worse than our method that utilizes offline data during online learning (especially on more challenging tasks like the Large and Ultra AntMaze environments in Figure 11 and on all HumanoidMaze tasks in Figure 16).
>
> In addition, our method is not a naive combination of SPiRL (Pertsch et al. (2021)) and ExPLORe (Li et al. (2024)). Pertsch et al. (2021) use a KL constraint between the high-level policy and a state-dependent prior obtained from offline pretraining. We found this design can actually hurt the online performance. We show that a simpler design without the KL constraint works much better. In Figure 7, we compare (Ours (KL)) with our method (Ours) and demonstrate that the final performance and the sample efficiency of the naive combination is much worse. As described in Appendix E, we borrow the policy parameterization from Haarnoja et al. (2018) and adopt a tanh policy parameterization with entropy regularization on the squashed space. Such design ensures that the online high-level policy is not explicitly constrained to a pre-trained prior, allowing the online policy to learn skill distributions that are more suitable for the task online. It also allows the addition of entropy regularization to the high level policy, which helps exploration.
>
> These careful designs are what make our method extremely stable, sample efficient, and scalable to more complex tasks. We show additional results on four new domains with two locomotion domains and two manipulation domains in Figure 3. We selected these environments from OGBench [1], since they provided challenging, long-horizon tasks which also require exploration to solve, making them well-suited for testing the use of offline data to accelerate online exploration. In the challenging HumanoidMaze domain, our method is often the only method that achieves non-zero success rate on the four most difficult mazes. On manipulation tasks, our method consistently outperforms all prior methods on all domains with the only exception on Scene where one of the baselines (Offline w/ HILP) performs better. It is worth noting that Offline w/ HILP is a novel baseline that we introduced to also leverage offline data twice, both during offline and online learning with the only difference being that the unsupervised skill pre-training algorithm is HILP (instead of using trajectory VAE). This further demonstrates that the principle of leveraging offline data for both skill pre-training and online learning is effective. The effectiveness of our method across seven domains further highlights the importance of a careful combination of skill pre-training and effective online learning that utilizes the offline data.
>
> [1] Park, Seohong, et al. "OGBench: Benchmarking Offline Goal-Conditioned RL." arXiv preprint arXiv:2410.20092 (2024).

---

> ### Author Response · Authors · 2024-11-21
> **Author Response (2/3)**
>
> **Limited discussion on HRL**
>
> Thanks for pointing out these related works that are really relevant to our work. We have added two new sections in the related works to discuss the relationship between our work and prior works in hierarchical RL and options framework (please see Section 2, “Hierarchical reinforcement learning” and “Options framework”). While some prior HRL methods simultaneously learn the low-level skill policies and the high-level policy online, others opt for a simpler formulation where the low-level skills are pre-trained offline and kept frozen during online learning. None of the prior HRL methods simultaneously leverage offline skill pre-training and offline data as additional off-policy data for high-level policy learning online. As we show in our experiments, both of them are crucial in enabling extremely sample efficient learning on challenging sparse-reward tasks, sometimes even solving tasks that all prior methods cannot (e.g., Figure 16 on HumanoidMaze).
>
> It is also worth noting that we have already discussed OPAL [1] in the unsupervised skill discovery section in the related work in our initial submission. Our unsupervised skill pre-training implementation closely follows the implementation of OPAL [1] and SPiRL [2] where the latent skills are extracted using a VAE. In addition, [3] does not use reinforcement learning and only focuses on extracting primitives, so we integrated this reference into our “unsupervised skill discovery” paragraph in the related work instead.
>
> [1] Ajay, Anurag, et al. "Opal: Offline primitive discovery for accelerating offline reinforcement learning." arXiv preprint arXiv:2010.13611 (2020).
>
> [2] Pertsch, Karl, Youngwoon Lee, and Joseph Lim. "Accelerating reinforcement learning with learned skill priors." Conference on robot learning. PMLR, 2021.
>
> [3] Paraschos, Alexandros, et al. "Probabilistic movement primitives." Advances in neural information processing systems 26 (2013).
>
> **High-level policy that is updated every 𝐻 timesteps and keeps the pre-trained skill and trajectory encoder fixed during the online phase. This limits the adaptability of the method.**
>
> While we acknowledge that this is indeed a limitation of our approach and using a more adaptive skill framework (e.g., options framework) can address this limitation (we add a new paragraph to discuss it in our related work section), our design (where the high-level policy outputs a skill at a regular interval) is a common design that appears in many prior methods [1-9] and many of them keep the skills fixed during online learning ([1-4]), and find it to be effective. In practice, we also find such design to be sufficient for a wide range of tasks (now with four new challenging domains in addition to the ones we tested in our initial submission).
>
> [1] Dalal, Murtaza, Deepak Pathak, and Russ R. Salakhutdinov. "Accelerating robotic reinforcement learning via parameterized action primitives." Advances in Neural Information Processing Systems 34 (2021): 21847-21859.
>
> [2] Gehring, Jonas, et al. "Hierarchical skills for efficient exploration." Advances in Neural Information Processing Systems 34 (2021): 11553-11564.
>
> [3] Pertsch, Karl, Youngwoon Lee, and Joseph Lim. "Accelerating reinforcement learning with learned skill priors." Conference on robot learning. PMLR, 2021.
>
> [4] Ajay, Anurag, et al. "Opal: Offline primitive discovery for accelerating offline reinforcement learning." arXiv preprint arXiv:2010.13611 (2020).
>
> [5] Xie, Kevin, et al. "Latent skill planning for exploration and transfer." arXiv preprint arXiv:2011.13897 (2020).
>
> [6] Gupta, Abhishek, et al. "Relay policy learning: Solving long-horizon tasks via imitation and reinforcement learning." arXiv preprint arXiv:1910.11956 (2019).
>
> [7] Fang, Kuan, et al. "Dynamics learning with cascaded variational inference for multi-step manipulation." arXiv preprint arXiv:1910.13395 (2019).
>
> [8] Merel, Josh, et al. "Neural probabilistic motor primitives for humanoid control." arXiv preprint arXiv:1811.11711 (2018).
>
> [9] Nachum, Ofir, et al. "Data-efficient hierarchical reinforcement learning." Advances in neural information processing systems 31 (2018).

---

> ### Author Response · Authors · 2024-11-21
> **Author Response (3/3)**
>
> **Dependence of Data Quality**
>
> To gain more insights on when we expect our method to work and the data quality dependency, we experimented on a narrow, expert dataset and three other datasets used in an offline goal-conditioned RL benchmark (OGBench [1]), shown in Figure 19. We picked datasets from this benchmark as it features a more diverse set of offline data distributions and hoped that it would provide more insights on when our method works or fails. As a summary, we find that our method does not need the dataset to be very diverse (e.g., collected by unsupervised RL agents). Completely exploratory dataset can actually break our method due to the lack of behavioral structure that can be extracted as skills. Our method excels at learning from datasets that contain segments of meaningful (e.g., navigating around the maze) behaviors. We discuss the results in detail below.
>
>
> The four datasets we consider are ordered in decreasing difficulty:
> - Expert: collected by a non-noisy expert policy that we train ourselves.
> - Navigate: collected by a noisy expert policy that randomly navigates the maze (from OGBench).
> - Stitch: collected by the same noisy expert policy but with much shorter trajectory length (from OGBench)
> - Explore: collected by moving the ant in random directions, where the direction is re-sampled every 10 environment steps. A large amount of action noise is also added (from OGBench).
>
>
> As expected, the baseline ExPLORe shows a gradual performance degradation from Expert to Navigate, to Stitch, and to Explore. All skill-based methods (including our method) fail completely on Explore. This is to be expected because the Explore dataset contains too much noise and the skills extracted from the dataset are likely very poor and meaningless. The high-level policy then would have trouble composing these bad skills to perform well in the environment. On Navigate and Stitch, our method outperforms other baselines, especially on the more challenging Stitch dataset where it is essential to stitch shorter trajectory segments together. On Expert, all methods perform similarly with ExPLORe doing slightly better. We hypothesize that this is because with the expert data, online learning does not require as much exploration, and skill-based methods are mostly beneficial when there is a need for structured exploratory behaviors.
>
>
> To further test our method’s ability to handle different offline data quality, in our original submission, we included ablation studies where the offline data is corrupted. We tested a dataset without transitions near the goal location (Insufficient Coverage), and a low-data setting where 95% of the trajectories are removed from the dataset (5% Data). While we see performance degradation from such data erasure, our method is still the most robust, consistently outperforming the baselines (Figure 20).
>
>
> We hope that these additional experiments provide more insights on when we expect our skill-based method to work or not work.
>
>
> [1] Park, Seohong, et al. "OGBench: Benchmarking Offline Goal-Conditioned RL." arXiv preprint arXiv:2410.20092 (2024).
>
>
> **Training stability in HRL**
>
> Many instability problems in HRL methods stem from the fact that both the low-level policy and the high-level policy are learning simultaneously at the same time. We circumvent such issues by pre-training low-level skill policies offline using a static dataset and then keep them fixed during online learning. In addition, our method utilizes the offline data as additional off-policy data for the high-level actor-critic RL agent such that the high-level RL agent can sample high-level transitions from offline data to perform TD updates. This effectively increases the amount of data that the high-level RL agent has access to right from the beginning of online learning, further stabilizing high-level policy learning.
>
> **Hope these address all of your questions and concerns. Thank you again for your time to review and if you have any remaining questions or concerns, please let us know!**

---

> > ### Comment · Reviewer_eVtj · 2024-11-29
> >
> > Thank the authors for the detailed reply, especially the data quality part. However, I will keep my score as I still have concerns about the paper's novelty.

---

> ### Author Response · Authors · 2024-11-29
>
> Thanks you again for your time to review our paper and read over our response. Could you be more specific about your concerns on the paper's novelty and how we did not sufficiently address your concerns in our response? We have showed with our experiments that our method outperforms prior approaches across **seven domains** and careful ablation to show that our key algorithm designs (e.g., using the offline data twice, removing the KL) are crucial in achieving the performance gain. We have also included a new sensitivity analysis experiments (Figure 5) that provides more insights to our algorithm. **If there are any new experiments or specific questions that could help improve the paper, please let us know!**

---

> ### Comment · Reviewer_eVtj · 2024-11-30
>
> Thanks for the quick reply. I appreciate the new experiments. Regarding the novelty, you mentioned the key insights are mainly i) using offline data twice and ii) removing the KL. For i) using the offline data twice, from the concept level, ExPLORe introduces using the offline data with optimistic rewards for better online learning. Built upon ExPLORe, the proposed method learns skills from offline data and shows better results, while the effectiveness of extracting skills is already demonstrated in many previous papers in both online and offline settings. For ii) removing the KL, it's more like an empirical observation (you included this in the practical implementation details section) rather than new insights. To make it a new insight for the community, you need to e.g., reveal the reason why removing KL is necessary and when (under which conditions) should we remove the KL, etc, which requires thorough analysis.

---

> > ### Author Response · Authors · 2024-11-30
> >
> > Thanks for your time in reviewing and for the quick response. Regarding the novelty, we would like to emphasize that **none of the prior works have effectively used offline data twice for both skill pre-training and online learning**. Our novelty lies in the observation that offline data can be simultaneously used for both skill pre-training and as additional off-policy data (which none of the prior works have observed), as well as the careful design choices that lead to strong performance in practice. ExPLORe only uses the offline data during online learning and does not perform pre-training, and prior skill-based methods did not even consider using offline data as additional prior data during online learning. In prior works, the usage of unlabeled offline data has been completely disjoint: either fully offline for skill pre-training or fully online as additional data for online learning. We show that precisely this combination is what differentiates our method in terms of performance. For example, in the HumanoidMaze environment, all of the prior methods completely failed on the more difficult Large and Giant mazes with near 0 success rate throughout training, but our method, with this combination, solves all of these tasks with high success rate (from 55\% to 80\%).
> >
> > In addition, we provided the necessary design details (e.g., removing the KL) that enables our method to achieve such sample efficiency, and we showed empirically that this design is beneficial. While our paper does not focus on these design details, we believe it is still a very valuable contribution to the community because **it, along with using offline data for both pre-training and as additional data for online RL, enables online RL to operate at a level of sample efficiency significantly better than previous state-of-the-art.**
> >
> > **We hope this addressed your concerns, and if there are any new experiments or specific questions that could help us improve the paper further, please let us know! Thanks again!**

---

### Author Response · Authors · 2024-11-21
**General Response**

We would like to thank all the reviewers for the insightful reviews and detailed feedback. We have responded to each reviewer individually for their specific concerns and questions and updated the submission PDF with the changes in blue color.


In case the reviewers would like to read about our responses to other reviewers and other new experiments we conducted, we provide a summary below.


**Concern 1: Limited novelty. The proposed method makes limited progress.**

First of all, our method is not just a naive combination of ExPLORe [1] and SPiRL [2]. We showed in Figure 7 that the naive combination (ours with KL) is worse in both sample efficiency and the final performance on AntMaze-Large.  SPiRL uses a KL constraint between the high-level policy and a state-dependent prior obtained from offline pretraining. We found the KL constraint to hurt performance. Our implementation removed the KL constraint and replaced it with entropy regularization on a squashed tanh space.

In addition, our method makes significant improvements over all prior methods. We now evaluate our method on 7 domains (42 tasks in total!), and show that we outperform all baselines (Figure 3) on each domain except Scene. On Scene, the baseline that outperforms our method is a baseline that we introduce that leverages offline data twice (one of the key ideas behind our method), both during offline and online learning. In addition, on the HumanMaze domain, our method is the only method that solves all tasks whereas all other methods fail almost completely.

These additional results demonstrate the progress that our method has made in making RL algorithms more effective at leveraging unlabeled offline data to accelerate online RL to the level that none of the prior methods were able to achieve.

[1] Qiyang Li, Jason Zhang, Dibya Ghosh, Amy Zhang, and Sergey Levine. Accelerating exploration with unlabeled prior data. Advances in Neural Information Processing Systems, 36, 2024.

[2] Pertsch, Karl, Youngwoon Lee, and Joseph Lim. "Accelerating reinforcement learning with learned skill priors." Conference on robot learning. PMLR, 2021.

**Concern 2: Ablation is lacking.**

We have added additional sensitive analysis experiments for our method. Figure 5 shows the performance of our method under different RND coefficient values (for optimistic labeling) and different skill horizon length. While it is crucial to use a non-zero RND coefficient value to encourage exploration, the performance of our method is robust to changes in RND coefficient values. We use a fixed RND coefficient for all our experiments across all the domains. We also use a fixed horizon length (H=4) for all our experiments across all the domains.


**Other New Experiments:**


*Ablation experiment where the ground truth reward is available in the offline dataset. How does our method compare to offline-to-online RL methods?  (Figure 8).*
- We found our method (with access to the ground truth reward), despite not designed for this setting, to even outperform SOTA offline-to-online RL methods on 4 AntMaze tasks and two of the three Kitchen tasks. This further highlights the effectiveness of our method.


*Experiments on D4RL Play datasets for AntMaze. Is the conclusion different from the Diverse datasets that we used in our initial submission? (Figure 15)*
- We found that on the Play datasets, our method also outperforms all baselines. The conclusion from the Play datasets is similar to the conclusion from the Diverse datasets.


*Experiments on datasets with different qualities. When do we expect our method to work/fail? (Figure 19)*
- We found that on an extremely exploratory dataset with largely random actions (plot on the left in Figure 19), our method is not able to extract meaningful skills, and fails to learn the task.
- Our method is the most suitable for datasets where there exists segments of meaningful behaviors (two middle subplots in Figure 19)
- Our method does not have an advantage over baselines on expert datasets (plot on the right in Figure 19).

**We would like to thank all the reviewers again for their time. If there are any remaining questions or concerns, please let us know! We would be happy to run any additional experiments that you think might improve our paper!**

---

### Meta-Review · Area_Chair_YL7y · 2024-12-22

**Metareview:**

This paper proposes using offline data to first extract a low-level skill policy $\pi(a|s, z)$, and then learn a high-level policy $\psi(a | s, z)$ that combines them during the online phase.

Strengths
1. Impressive gains over several baselines in Figure 3.
2. Authors added new results that show gains over offline-to-online RL approaches in Figure 8 which is a common approach.

Weakness:
1. The approach doesn't introduce a major idea although I would say it skillfully uses existing ideas.

2. Most of the experiments are on state-based observations and many of them are grid-based. For a paper for ICLR 2025, I think the domains to be addressed should be closer to real-world challenges especially given the progress in the ML community at large.

3. The approach depends on the quality of the data. While offline data is unlabeled, meaning there are no rewards, which is nice compared to offline RL approaches, it still requires access to semantically meaningful trajectories from which skills can be extracted. E.g., if the trajectories are random walks, then such an approach wouldn't work. Where can we expect to find such data? If the approach used something such as video data which are more abundantly available, then it would make the approach more practical.

Overall, I think this paper uses existing ideas in RL to make impressive improvements over a variety of common benchmarks. The main concern is a mismatch with real-world problems where high-quality datasets may be hard to get. Further, the experiments are not on real-world problems which matters more for a paper where the core contribution is a skillful combination of existing ideas. One way to address this is to use more visual environments. Authors have also added a lot of new experiments during the rebuttal period that need more scrutiny. E.g., Figure 8 compares offline-to-online RL approaches to the proposed approach, however, the proposed approach requires high-quality data while offline RL can work with low-quality and high-coverage data. Further, it is not clear to me how authors claim gains over Cal-QL in 6/7 domains in Figure 8. For both medium play and medium diverse both cal-ql (green) and the proposed approach (black) achieve a return of 1. Also, the green plots are truncated in some places as they are read from the previous papers. For a fair comparison, all results should have the same number of steps.

For now, I am recommending a rejection with strong encouragement to submit again with more experiments and add more clarity for the results.

**Additional Comments On Reviewer Discussion:**

Reviewers raised the following main concerns:

1. Lack of novelty (reviewer t68W): Authors addressed that their approach isn't a trivial combination of ExPLORe and SPiRL and demonstrate that the naive combination doesn't work as well. Specifically, the authors claim that no past work uses the offline data twice -- once for learning low-level policy during pre-training and a second time during the online phase. I think this is a less important point and I have discounted the novelty concerns. However, I was expecting experiments in more realistic environments.

2. Lack of comparison with baselines and domains: Authors have added a comparison with offline-to-online approaches in Figure 8 although I don't know how they read success from this. Authors have also added other ablations such as the RND coefficient.

3. Concerns that the dataset should be of high quality: Authors agree that when the data is low-quality then their approach doesn't work as one would expect from looking at the approach. I think this limitation is fine provided authors can justify more where they would hope to get such data in practice. I don't think authors tackle this problem as they mostly rely on existing simulated benchmarks.

Overall, my main addressable concerns are experiments being unclear (Figure 8) and the lack of more realistic problems (e.g., including more domains from OGBench can help). Second is that the approach relies on data being of high quality. It would be nice to show a comparison with offline-to-online approaches over different qualities of datasets, and/or argue where this dataset can be found in practice.

---

### Decision · Program_Chairs · 2025-01-22

Reject